

# Lack of symmetry restoration after a quantum quench: An entanglement asymmetry study

**Filiberto Ares[1], Sara Murciano[2,3], Eric Vernier[4] and Pasquale Calabrese[1,5]**

**1** SISSA and INFN Sezione di Trieste, via Bonomea 265, 34136 Trieste, Italy
**2** Walter Burke Institute for Theoretical Physics, Caltech, Pasadena, CA 91125, USA
**3** Department of Physics and IQIM, Caltech, Pasadena, CA 91125, USA
**4** Laboratoire de Probabilités, Statistique et Modélisation,
CNRS - Univ. Paris Cité - Sorbonne Univ., 75013 Paris, France
**5** International Centre for Theoretical Physics (ICTP),
Strada Costiera 11, 34151 Trieste, Italy

## Abstract

We consider the quantum quench in the XX spin chain starting from a tilted Néel state which explicitly breaks the $U(1)$ symmetry of the post-quench Hamiltonian. Very surprisingly, the $U(1)$ symmetry is not restored at large time because of the activation of a non-Abelian set of charges which all break it. The breaking of the symmetry can be effectively and quantitatively characterised by the recently introduced entanglement asymmetry. By a combination of exact calculations and quasi-particle picture arguments, we are able to exactly describe the behaviour of the asymmetry at any time after the quench. Furthermore we show that the stationary behaviour is completely captured by a non-Abelian generalised Gibbs ensemble. While our computations have been performed for a non-interacting spin chain, we expect similar results to hold for the integrable interacting case as well because of the presence of non-Abelian charges also in that case.

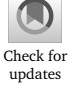

# 1   Introduction

Although the charm of nature resides in the presence of symmetries, lots of interesting and relevant phenomena are due to their breaking. This can happen spontaneously when, despite the corresponding conservation laws are respected by the theory, the state of the system is not symmetric, or explicitly, when the Hamiltonian that dictates the dynamics of the system contains terms that do not respect the symmetry. Much work has been done about many aspects of symmetry breaking in different branches of physics. However, little attention has been paid to study the non-equilibrium dynamics of a broken symmetry in a quantum many-body system; for example, how is the time evolution of a global symmetry if the system is initiated in a state that breaks it and then it is let evolve with a Hamiltonian that does preserve it. Regarding this, an important point is if the symmetry can be dynamically restored and how fast it does. Only some few works have analysed this question in spin chains both for a global $U(1)$ symmetry [1,2] and for a discrete $\mathbb{Z}_2$ group [3,4] using spin correlators.

The absence of studies on this problem is perhaps due to the lack of a proper quantity that measures how much a symmetry is broken. In extended quantum systems, this issue is intrinsically bound to consider a specific subsystem. In the recent Ref. [5], a subsystem measure of symmetry breaking, dubbed *entanglement asymmetry*, has been introduced by employing tools from the theory of entanglement in quantum many-body systems. In such work, the new entanglement asymmetry is applied to study the time evolution of a $U(1)$ symmetry in a spin-1/2 chain initiated in a tilted ferromagnetic configuration, which breaks that symmetry, after a sudden global quench to the XX spin chain Hamiltonian, which respects it. The analysis of the entanglement asymmetry reveals not only that the symmetry is restored but also that the more the symmetry is initially broken, the smaller is the time necessary to recover it. This surprising and counterintuitive phenomenon is a sort of quantum version of the still controversial Mpemba effect [6,7]— more the system is initially out of equilibrium, the faster it relaxes.

The present work is a complement of the analysis done in Ref. [5]. Here we also use the entanglement asymmetry to study the dynamics of the same $U(1)$ symmetry after a quench to the XX Hamiltonian, but preparing the spin chain in a tilted Néel configuration instead of the tilted ferromagnetic one. This change in the initial state of the quench protocol drastically

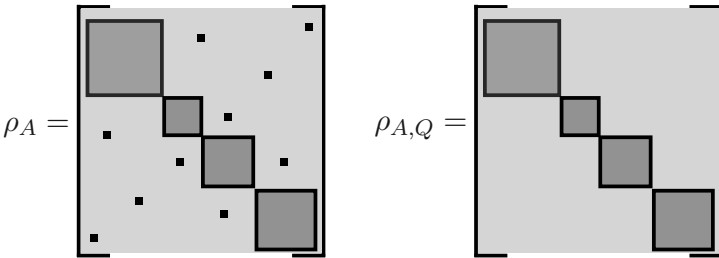

Figure 1: Schematic comparison of the form of the density matrices $\rho_A$ and $\rho_{A,Q}$ in the eigenbasis of the subsystem charge $Q_A$. If the full system has not a definite charge, the reduced density matrix $\rho_A$ contains both non-zero diagonal and off-diagonal blocks. By projecting over all the symmetry sectors, we obtain $\rho_{A,Q}$, where the off-diagonal blocks are annihilated. The difference $\Delta S_A^{(n)}$ between the entanglement entropies of these matrices gives the entanglement asymmetry (1).

modifies the time evolution of the $U(1)$ symmetry, which now is not restored at large times.

**Entanglement asymmetry:** The setup we are interested in is an extended quantum system in a pure state $|\Psi\rangle$, defined in a bipartite Hilbert space $\mathcal{H} = \mathcal{H}_A \otimes \mathcal{H}_B$, where $\mathcal{H}_A$ and $\mathcal{H}_B$ are respectively associated to two spatial regions $A$ and $B$. The state of $A$ is described by the reduced density matrix $\rho_A$ obtained by taking the partial trace to the complementary subsystem as $\rho_A = \text{Tr}_B(|\Psi\rangle\langle\Psi|)$. We further consider a charge operator $Q$ with integer eigenvalues that generates a $U(1)$ symmetry group. For the given bipartition, $Q$ is the sum of the charge in each region, $Q = Q_A + Q_B$. If $|\Psi\rangle$ is an eigenstate of $Q$, then $[\rho_A, Q_A] = 0$ and $\rho_A$ displays a block-diagonal structure in the charge sectors of $Q_A$. On the other hand, if the state $|\Psi\rangle$ breaks the symmetry, then $[\rho_A, Q_A] \neq 0$ and $\rho_A$ is not block-diagonal in the eigenbasis of $Q_A$. Based on these observations, the entanglement asymmetry in the subsystem $A$ is defined as

$$\Delta S_A = S(\rho_{A,Q}) - S(\rho_A), \tag{1}$$

where

$$S(\rho) = -\text{Tr}(\rho \log \rho) \tag{2}$$

is the von Neumann entropy associated to the density matrix $\rho$. The matrix $\rho_{A,Q}$ is obtained from $\rho_A$ by projecting over all the symmetry sectors of $Q_A$ such that $\rho_{A,Q} = \sum_{q \in \mathbb{Z}} \Pi_q \rho_A \Pi_q$, where $\Pi_q$ is the projector onto the eigenspace of $Q_A$ with charge $q \in \mathbb{Z}$. Thus $\rho_{A,Q}$ is block-diagonal in the eigenbasis of $Q_A$. In Fig. 1, we schematically represent the form of $\rho_A$ and $\rho_{A,Q}$.

As a measure of symmetry breaking, the entanglement asymmetry (1) satisfies two fundamental properties. First, it is non-negative, $\Delta S_A \geq 0$, since it can be rewritten as the relative entropy between $\rho_A$ and $\rho_{A,Q}$, $\Delta S_A = \text{Tr}[\rho_A(\log \rho_A - \log \rho_{A,Q})]$, which by definition can never be negative [8]. Second, it vanishes, $\Delta S_A = 0$, iff the state of subsystem $A$ respects the symmetry associated to $Q$, i.e. $[\rho_A, Q_A] = 0$. In fact, in this case, $\rho_A$ is block diagonal in the eigenbasis of $Q_A$, the projector $\Pi_q$ leaves it invariant and, therefore, $\rho_{A,Q} = \rho_A$. When this occurs, the entanglement entropy $S(\rho_A)$, a key quantity in the study of quantum many-body systems that measures the degree of entanglement between subsystems $A$ and $B$, can be resolved into the contribution of each charge sector [9–11]. This fact has recently motivated an intense research activity on the interplay between entanglement and symmetries both theoretically [12–17] and experimentally [18–22].

**Replica trick:** The definition of Eq. (1) makes clear the connection between the entanglement asymmetry and the entanglement entropy. For this reason, in order to investigate the asymmetry, we can apply many of the techniques developed for the analysis of the (symmetry-resolved) entanglement entropy in the bipartite setup described above; in particular, one of the most fruitful is the *replica trick* [23, 24]. If we introduce the Rényi entropies,

$$S^{(n)}(\rho) = \frac{1}{1-n} \log \mathrm{Tr}(\rho^n),\tag{3}$$

then the von Neumann entropy (2) can be obtained as the limit $n \to 1$ of Eq. (3). With this in mind, the definition of Eq. (1) can be extended by replacing the von Neumann entropy $S(\rho)$ with the Rényi entropy $S^{(n)}(\rho)$,

$$\Delta S_A^{(n)} = S^{(n)}(\rho_{A,Q}) - S^{(n)}(\rho_A).\tag{4}$$

The main advantage of the Rényi entanglement asymmetry is that it is easier to calculate for integer $n$, and Eq. (1) can be recovered by taking the limit $\lim_{n\to 1} \Delta S_A^{(n)} = \Delta S_A$. Moreover, for integer $n \geq 2$, it can be experimentally accessed via randomised measurements in ion-trap setups [25–28]. As in the case $n \to 1$, $\Delta S_A^{(n)}$ is always non-negative [29] and vanishes if and only if $\rho_A$ respects the symmetry, i.e. $[\rho_A, Q_A] = 0$.

**Quench protocol:** As we already announced, the goal of this paper is to use the entanglement asymmetry to analyse in a subsystem of an infinite spin-1/2 chain the non-equilibrium dynamics of a broken $U(1)$ symmetry after a global quantum quench. In particular, we take the transverse magnetisation,

$$Q = \frac{1}{2} \sum_j \sigma_j^z,\tag{5}$$

as the charge that generates the $U(1)$ symmetry and as subsystem $A$ a set of $\ell$ contiguous spins. The specific quench protocol that we consider is the following. We initially prepare the spin chain in the cat state

$$|\Psi(0)\rangle = \frac{|\mathrm{N}, \theta\rangle - |\mathrm{N}, -\theta\rangle}{\sqrt{2}},\tag{6}$$

where $|\mathrm{N}, \theta\rangle$ denotes the tilted Néel state,

$$|\mathrm{N}, \theta\rangle = e^{i\frac{\theta}{2}\sum_j \sigma_j^y} |\downarrow\uparrow\cdots\rangle,\tag{7}$$

which breaks the $U(1)$ symmetry associated to the transverse magnetisation (5). We take as initial configuration the linear combination of Eq. (6), instead of the state (7), because the corresponding reduced density matrix $\rho_A$ is Gaussian, a property that makes much easier both the numerical and analytical study of the entanglement asymmetry (see Appendix A for a proof of the Gaussianity of $\rho_A$ for the state $|\Psi(0)\rangle$). Then we evolve it in time

$$|\Psi(t)\rangle = e^{-itH} |\Psi(0)\rangle,\tag{8}$$

with the Hamiltonian of the XX spin chain

$$H = -\frac{1}{4} \sum_{j=-\infty}^{\infty} \left[ \sigma_j^x \sigma_{j+1}^x + \sigma_j^y \sigma_{j+1}^y \right],\tag{9}$$

which does preserve the charge (5), i.e. $[H, Q] = 0$.

The tilting angle $\theta \in [0, \pi]$ tunes how much the symmetry is initially broken. In fact, the tilted Néel state breaks the $U(1)$ symmetry associated to the charge (5) when $\theta \neq 0, \pi$. On

the other hand, for $\theta = 0, \pi$, the transverse magnetisation has a defined value since all the spins point in the $z$-direction. Therefore, the initial entanglement asymmetry is $\Delta S_A^{(n)} = 0$ when $\theta = 0, \pi$, and $\Delta S_A^{(n)} > 0$ otherwise. In particular, the symmetry is maximally broken for $\theta = \pi/2$, when all the spins are aligned in the $x$-direction, and $\Delta S_A^{(n)}$ reaches its maximum value.

The rest of the paper will be devoted to study the time evolution of the Rényi entanglement asymmetry after the quench of Eq. (8). By applying the quasi-particle picture of entanglement, we derive an exact analytic expression for $\Delta S_A^{(n)}(t)$ in the scaling limit $t, \ell \to \infty$ with $\zeta = t/\ell$ finite. We obtain that, at large times after the quench, the entanglement asymmetry tends to a non-zero constant value, except at $\theta = \pi/2$, for which it does go to zero. This implies that, after the quench (8), the $U(1)$ symmetry is not restored unless it is maximally broken at time zero.

Using the generalised Gibbs ensemble description of the post-quench relaxation, we will show that the reason behind the lack of symmetry restoration is the existence of a set of charges of the evolution Hamiltonian (9) that do not commute with $Q$ and whose initial state expectation value is not zero due to the breaking of translational invariance of the tilted Néel state.

**Outline:** The paper is organised as follows. In Sec. 2, we describe the general approach to compute the Rényi entanglement asymmetry (4) in terms of a generalised version of the charged moments of the reduced density matrix, and we discuss how to efficiently calculate the latter when the reduced density matrix is a fermionic Gaussian state. In this section, we also review the results of Ref. [5] for the entanglement asymmetry when the initial configuration is the tilted ferromagnetic state. In Sec. 3, we obtain the exact time evolution of the charged moments when the spin chain is quenched from the cat tilted Néel state. With this result, in Sec. 4, we analyse the behaviour of the Rényi entanglement asymmetry after the quench, finding that the initially broken $U(1)$ is generally not restored at large times. We check the analytical expressions derived in these sections with exact numerical calculations. In Sec. 5, we explain the lack of symmetry restoration in terms of the generalised Gibbs ensemble that describes the post-quench stationary behaviour. We conclude in Sec. 6 with some remarks and future prospects. We include several appendices where we derive and discuss with more details some of the results of the main text.

## 2 Entanglement asymmetry and charged moments

In this section, we introduce the basic tools that we employ to calculate the Rényi entanglement asymmetry defined in Eq. (4). We also review the known results for the quench from the tilted ferromagnetic state, following Ref. [5].

### 2.1 Charged moments

In order to evaluate Eq. (4), we consider the Fourier representation of the projector $\Pi_q$. Then the projected density matrix $\rho_{A,Q}$ can be re-expressed in the form

$$\rho_{A,Q} = \int_{-\pi}^{\pi} \frac{d\alpha}{2\pi} e^{-i\alpha Q_A} \rho_A e^{i\alpha Q_A},\tag{10}$$

and its moments as

$$\text{Tr}(\rho_{A,Q}^n) = \int_{-\pi}^{\pi} \frac{d\alpha_1 \dots d\alpha_n}{(2\pi)^n} Z_n(\boldsymbol{\alpha}),\tag{11}$$

where $\boldsymbol{\alpha} = \{\alpha_1, \dots, \alpha_n\}$ and

$$Z_n(\boldsymbol{\alpha}) = \text{Tr}\left[\prod_{j=1}^{n} \rho_A e^{i\alpha_{j,j+1}Q_A}\right], \tag{12}$$

with $\alpha_{ij} \equiv \alpha_i - \alpha_j$ and $\alpha_{n+1} = \alpha_1$. From the previous expressions, it is straightforward to see that, if $[\rho_A, Q_A] = 0$, then $Z_n(\boldsymbol{\alpha}) = Z_n(\mathbf{0})$, which implies $\text{Tr}(\rho_{A,Q}^n) = \text{Tr}(\rho_A^n)$ and $\Delta S_A^{(n)} = 0$, consistently with the properties of the asymmetry that we discussed in the introduction. We call the objects $Z_n(\boldsymbol{\alpha})$ charged moments since they can be seen as a non-trivial generalisation of the ones considered in the study of symmetry-resolved entanglement [10].

If $\rho_A$ and $e^{i\alpha Q_A}$ are Gaussian, i.e. they are the exponential of a quadratic fermionic operator, then the charged moments $Z_n(\boldsymbol{\alpha})$ can be obtained from the fermionic two-point correlations. After a Jordan-Wigner transformation, the XX spin chain Hamiltonian of Eq. (9) is quadratic in terms of the fermionic operators $\boldsymbol{c}_j = (c_j, c_j^\dagger)$, which satisfy the canonical anticommutation relations $\{c_j, c_{j'}^\dagger\} = \delta_{j,j'}$. Then it can be diagonalised by performing a Fourier transformation to momentum space [30]. The one-particle dispersion relation is $\epsilon_k = -\cos(k)$. Therefore, if the initial state $|\Psi(0)\rangle$ satisfies the Wick theorem, then the reduced density matrix $\rho_A(t) = \text{Tr}_B |\Psi(t)\rangle\langle\Psi(t)|$ after the quench (8) is a Gaussian operator for all values of $t$ and it can be obtained from the time-dependent two-point correlation matrix restricted to subsystem $A$ [31]

$$\Gamma_{jj'}(t) = 2\langle\Psi(t)| \boldsymbol{c}_j^\dagger \boldsymbol{c}_{j'} |\Psi(t)\rangle - \delta_{j,j'}, \quad j, j' \in A. \tag{13}$$

If the subsystem $A$ is a single interval of $\ell$ contiguous sites, then $\Gamma(t)$ has dimension $2\ell \times 2\ell$.

In terms of the fermionic operators $\boldsymbol{c}_j$, the transverse magnetisation (5) is related to the fermionic number operator, $Q = \sum_j (c_j^\dagger c_j - 1/2)$, which is also quadratic. Therefore, Eq. (12) is the trace of the product of Gaussian fermionic operators, $\rho_A(t)$ and $e^{i\alpha_{j,j+1}Q_A}$. Applying the composition rules derived in Refs. [32, 33] for the trace of a product of Gaussian operators, see also Appendix B, we can calculate the charged moments of Eq. (12) from the correlation matrix $\Gamma(t)$ with the formula [5]

$$Z_n(\boldsymbol{\alpha}, t) = \sqrt{\det\left[\left(\frac{I - \Gamma(t)}{2}\right)^n \left(I + \prod_{j=1}^{n} W_j(t)\right)\right]}, \tag{14}$$

where $W_j(t) = (I + \Gamma(t))(I - \Gamma(t))^{-1} e^{i\alpha_{j,j+1}n_A}$ and $n_A$ is a diagonal matrix with $(n_A)_{2j,2j} = 1$, $(n_A)_{2j-1,2j-1} = -1$, $j = 1, \cdots, \ell$. We will use the result in Eq. (14) to exactly compute the time evolution of $\Delta S_A^{(n)}(t)$ and verify the analytical predictions that we find throughout the manuscript.

## 2.2 Quench from the tilted Ferromagnetic state

It is illustrative to review what happens when in a global quantum quench to the XX Hamiltonian the chain is initiated in the cat state of Eq. (6) but building it with the tilted ferromagnetic state,

$$|\text{F}, \theta\rangle = e^{i\frac{\theta}{2}\sum_j \sigma_j^y}|\uparrow\uparrow\cdots\rangle, \tag{15}$$

instead of the tilted Néel configuration. This case was studied in Ref. [5] employing the formalism described above. In order to compare it with our findings for the tilted Néel case, it will be enough to present the exact time evolution after the quench of the charged moments (12) in the scaling limit $t, \ell \to \infty$ with $\zeta = t/\ell$ fixed,

$$Z_n(\boldsymbol{\alpha}, t) = Z_n(\mathbf{0}, t)e^{\ell(A_n(\boldsymbol{\alpha}) + B_n(\boldsymbol{\alpha}, \zeta))}, \tag{16}$$

where the functions $A_n(\boldsymbol{\alpha})$ and $B_n(\boldsymbol{\alpha}, \zeta)$ read, respectively,

$$A_n(\boldsymbol{\alpha}) = \int_0^{2\pi} \frac{dk}{2\pi} \log \prod_{j=1}^n f_k^{\mathrm{F}}(\theta, \alpha_{j,j+1}),$$

$$B_n(\boldsymbol{\alpha}, \zeta) = -\int_0^{2\pi} \frac{dk}{2\pi} \min(2\zeta|\epsilon_k'|, 1) \log \prod_{j=1}^n f_k^{\mathrm{F}}(\theta, \alpha_{j,j+1}), \tag{17}$$

and $f_k^{\mathrm{F}}(\theta, \alpha)$ is defined as

$$f_k^{\mathrm{F}}(\theta, \alpha) = i e^{i\Delta_k(\theta)} \sin\left(\frac{\alpha}{2}\right) + \cos\left(\frac{\alpha}{2}\right), \tag{18}$$

with

$$e^{i\Delta_k(\theta)} = \frac{2\cos\theta - (1 + \cos^2\theta)\cos k + i \sin^2(\theta)\sin(k)}{1 - 2\cos(\theta)\cos(k) + \cos^2\theta}. \tag{19}$$

This result has been obtained by combining two ingredients: the first one is that in this quench protocol, the $U(1)$ symmetry is restored in the large time limit [1, 2], and $B_n(\boldsymbol{\alpha}, \zeta) \to -A_n(\boldsymbol{\alpha})$ as $\zeta \to \infty$ such that $\Delta S_A^{(n)}(t) \to 0$. The second one consists in adapting the quasi-particle picture of entanglement in order to reconstruct the behaviour of $B_n(\boldsymbol{\alpha}, \zeta)$ for finite $\zeta$. Taking the Fourier transform (11) of Eq. (16), we obtain the result for $\Delta S_A^{(n)}(t)$. In particular, at time $t = 0$ and subsystem size $\ell \to \infty$, we find by applying the same saddle point method described in Sec. 4 for the cat tilted Néel state that

$$\Delta S_A^{(n)}(t=0) = \frac{1}{2}\log\ell + \frac{1}{2}\log\frac{\pi n^{\frac{1}{n-1}}\sin^2\theta}{8} + O(\ell^{-1}). \tag{20}$$

The limit $\theta \to 0$ is not well defined in the expression above and, in order to recover it, one should carefully consider $\theta \to 0$ and then the large interval regime. It is also important to mention that the $O(\ell^0)$ term in the asymptotic behaviour (20) for the cat tilted ferromagnet is slightly different than for the non-cat state $|\mathrm{F}, \theta\rangle$, which was obtained in Ref. [5]. About the dynamics of $\Delta S_A^{(n)}(t)$, we found that this quantity vanishes for large times as $t^{-3}$ for any value of $\theta$. Another feature, which follows from having a space-time scaling, is that larger subsystems require more time to recover the symmetry. Finally, we have observed the very odd and unexpected feature that the more the symmetry is initially broken, i.e. the larger $\theta$, the smaller the time to restore it, the aforementioned quantum Mpemba effect.

## 3 Charged moments after the quench from the tilted Néel state

In this section, we study the time evolution of the charged moments $Z_n(\boldsymbol{\alpha}, t)$ defined in Eq. (12) after the quench (8) from the tilted Néel state. To this end, we employ the determinant formula of Eq. (14) in terms of the fermionic two-point correlation matrix (13). Therefore, in the first part of this section, we calculate the latter in our specific quench. We then introduce some useful properties of determinants involving products of block Toeplitz matrices and their inverse. Finally, with these results and the quasi-particle picture of entanglement, we derive an exact analytic expression for the evolution $Z_n(\boldsymbol{\alpha}, t)$ after the quench.

### 3.1 Correlation functions

Since the tilted Néel state (7) is invariant by two-site translations, it is useful to rearrange the entries of the two-point correlation matrix $\Gamma(t)$ in $4 \times 4$ blocks of the form

$$\Gamma_{ll'}(t) = 2\left\langle \Psi(t) \left| \begin{pmatrix} c_{2l-1}^\dagger \\ c_{2l}^\dagger \\ c_{2l-1} \\ c_{2l} \end{pmatrix} (c_{2l'-1}, c_{2l'}, c_{2l'-1}^\dagger c_{2l'}^\dagger) \right| \Psi(t) \right\rangle - \delta_{l,l'}, \quad l,l' = 1,\ldots,\ell/2. \quad (21)$$

In this way, for an infinite spin chain, the correlation matrix $\Gamma(t)$ at time zero can be cast as a block Toeplitz matrix,

$$\Gamma_{ll'}(0) = \int_0^{2\pi} \frac{dk}{2\pi} \mathcal{G}_0(k,\theta) e^{-ik(l-l')}, \qquad l,l' = 1,\ldots \ell/2, \quad (22)$$

where the symbol $\mathcal{G}_0(k,\theta)$ is the $4 \times 4$ matrix

$$\mathcal{G}_0(k,\theta) = \begin{pmatrix} g_{11}(k,\theta) & e^{ik/2}g_{12}(k,\theta) & -if_{11}(k,\theta) & -ie^{ik/2}f_{12}(k,\theta) \\ e^{-ik/2}g_{12}(k,\theta) & -g_{11}(k,\theta) & -ie^{-ik/2}f_{12}(k,\theta) & if_{11}(k,\theta) \\ if_{11}(k,\theta) & ie^{ik/2}f_{12}(k,\theta) & -g_{11}(k,\theta) & -e^{ik/2}g_{12}(k,\theta) \\ ie^{-ik/2}f_{12}(k,\theta) & -if_{11}(k,\theta) & -e^{-ik/2}g_{12}(k,\theta) & g_{11}(k,\theta) \end{pmatrix}, \quad (23)$$

whose entries are given by

$$g_{11}(k,\theta) = -\cos(\theta) - \frac{\cos\theta \sin^2\theta(\cos k + \cos^2\theta)}{(1 + 2\cos k \cos^2\theta + \cos^4\theta)}, \quad (24)$$

$$g_{12}(k,\theta) = -\frac{\cos(k/2)(1-\cos^4\theta)}{1 + 2\cos k \cos^2\theta + \cos^4\theta}, \quad (25)$$

$$f_{11}(k,\theta) = -\frac{\cos\theta \sin^2\theta \sin k}{1 + 2\cos k \cos^2\theta + \cos^4\theta}, \quad (26)$$

$$f_{12}(k,\theta) = -\frac{\sin(k/2)\sin^4\theta}{1 + 2\cos k \cos^2\theta + \cos^4\theta}. \quad (27)$$

We refer the reader to Appendix C for a detailed derivation of this correlation matrix. As we mentioned before, the $U(1)$ symmetry generated by the transverse magnetisation corresponds to particle number conservation in fermionic language. This implies that the correlations $\langle \Psi(t)| c_j^\dagger c_{j'}^\dagger |\Psi(t)\rangle$ and $\langle \Psi(t)| c_j c_{j'} |\Psi(t)\rangle$ vanish when the symmetry is not broken, as actually happens for $\theta = 0, \pi$ at time zero.

After the quench to the XX spin chain, the correlation matrix $\Gamma(t)$ is also block Toeplitz, given that the post-quench Hamiltonian (9) is translationally invariant. In this case, it is useful to study separately the correlation functions $\langle \Psi(t)| c_j^\dagger c_{j'} |\Psi(t)\rangle$ and $\langle \Psi(t)| c_j^\dagger c_{j'}^\dagger |\Psi(t)\rangle$. For the former, we find in Appendix C that

$$\langle \Psi(t)| \begin{pmatrix} c_{2l-1}^\dagger \\ c_{2l}^\dagger \end{pmatrix} (c_{2l'-1}, c_{2l'}) |\Psi(t)\rangle = \frac{\delta_{ll'}}{2} + \int_0^{2\pi} \frac{dk}{4\pi} \mathcal{C}_t(k,\theta) e^{-ik(l-l')}, \quad (28)$$

where the symbol is provided by

$$\mathcal{C}_t(k,\theta) = \begin{pmatrix} \cos(2t\epsilon_{k/2})g_{11}(k,\theta) & e^{ik/2}(g_{12}(k,\theta) + i\sin(2t\epsilon_{k/2})g_{11}(k,\theta)) \\ e^{-ik/2}(g_{12}(k,\theta) - i\sin(2t\epsilon_{k/2})g_{11}(k,\theta)) & -\cos(2t\epsilon_{k/2})g_{11}(k,\theta) \end{pmatrix}. \quad (29)$$

On the other hand, the terms involving the correlation functions that vanish when the symmetry is respected, $\langle \Psi(t)| c_j^\dagger c_{j'}^\dagger |\Psi(t)\rangle$, are described by, see also Appendix C,

$$\langle \Psi(t)| \begin{pmatrix} c_{2l-1} \\ c_{2l} \end{pmatrix} (c_{2l'-1}, c_{2l'}) |\Psi(t)\rangle = \int_{-\pi}^{\pi} \frac{dk}{4\pi} \mathcal{F}_t(k,\theta) e^{-ik(l-l')}, \quad (30)$$

with

$$\mathcal{F}_t(k,\theta) = \begin{pmatrix} if_{11}(k,\theta) - f_{12}(k,\theta)\sin(2t\epsilon_{k/2}) & ie^{ik/2}\cos(2t\epsilon_{k/2})f_{12}(k,\theta) \\ ie^{-ik/2}\cos(2t\epsilon_{k/2})f_{12}(k,\theta) & -if_{11}(k,\theta) - f_{12}(k,\theta)\sin(2t\epsilon_{k/2}) \end{pmatrix}. \quad (31)$$

Now, combining Eqs. (28) and (30), we obtain the expression for the full correlation matrix as a function of time $t$,

$$\Gamma_{ll'}(t) = \int_{-\pi}^{\pi} \frac{dk}{2\pi} e^{-ik(l-l')} \mathcal{G}_t(k,\theta), \quad (32)$$

where

$$\mathcal{G}_t(k,\theta) = \begin{pmatrix} \mathcal{C}_t(k,\theta) & \mathcal{F}_t(k,\theta)^\dagger \\ \mathcal{F}_t(k,\theta) & -\mathcal{C}_t(-k,\theta)^* \end{pmatrix}. \quad (33)$$

As we explain in Appendix C, due to the rearrangement of the entries of $\Gamma(t)$ as a block Toeplitz matrix to adapt to the two-site periodicity of the initial state $|\Psi(0)\rangle$, the Brillouin zone of the post-quench Hamiltonian, which is fully translationally invariant, is halved and its dispersion relation appears in Eqs. (29) and (31) as $\epsilon_{k/2}$ instead of $\epsilon_k$.

To derive the stationary value of the charged moments and of the entanglement asymmetry at large times, we can average the time dependent terms in the symbol $\mathcal{G}_t(k,\theta)$ of $\Gamma(t)$. At $t \to \infty$, the functions $\sin(2t\epsilon_{k/2})$ and $\cos(2t\epsilon_{k/2})$ in Eq. (33) average to zero and the symbol simplifies,

$$\mathcal{G}_{t\to\infty}(k,\theta) = \begin{pmatrix} 0 & e^{ik/2}g_{12}(k,\theta) & -if_{11}(k,\theta) & 0 \\ e^{-ik/2}g_{12}(k,\theta) & 0 & 0 & if_{11}(k,\theta) \\ if_{11}(k,\theta) & 0 & 0 & -e^{ik/2}g_{12}(k,\theta) \\ 0 & -if_{11}(k,\theta) & -e^{-ik/2}g_{12}(k,\theta) & 0 \end{pmatrix}. \quad (34)$$

Note that, when we take the time average, some of the correlation functions $\langle\Psi(t)|c_j c_{j'}|\Psi(t)\rangle$ and $\langle\Psi(t)|c_j^\dagger c_{j'}^\dagger|\Psi(t)\rangle$ do not vanish for $\theta \neq 0, \pi/2$ and $\pi$. This is the first indicator that, in such case, the broken symmetry is not restored at large times after the quench, as we will see in the following sections. This is the main difference with respect to the quench from the tilted ferromagnetic state reviewed in Sec. 2.2.

## 3.2 Useful properties of block Toeplitz matrices

Before proceeding, we report two important properties of block Toeplitz matrices that will be useful to calculate the charged moments from Eq. (14). The determinant of that expression involves the product of the block Toeplitz matrices $(I + \Gamma(t))e^{i\alpha_{j,j+1}n_A}$ as well as the inverse matrix $(I - \Gamma(t))^{-1}$, which do not commute. In general, the latter is not block Toeplitz, and the same occurs with the product of block Toeplitz matrices. Therefore, we cannot in principle apply the well-known results on the asymptotic behaviour of block Toeplitz matrices, e.g. the Widom-Szegő theorem [34] or the Fisher-Hartwig conjecture [35–37], usually employed to study the entanglement entropy and other quantities in free fermionic systems. However, we formulate the following conjectures on the asymptotics of the determinant of a product of block Toeplitz matrices that may also contain the inverse of block Toeplitz matrices.

Let us denote by $T_\ell[g]$ a block Toeplitz matrix of dimension $d \cdot \ell$ with symbol the $d \times d$ matrix $g(k)$ defined on $k \in [0, 2\pi)$. That is, the entries of $T_\ell[g]$ are the Fourier coefficients of $g(k)$,

$$(T_\ell[g])_{ll'} = \int_0^{2\pi} \frac{dk}{2\pi} e^{-ik(l-l')} g(k), \quad l, l' = 1, \dots, \ell. \quad (35)$$

If we consider the product of $n$ different block Toeplitz matrices $T_\ell[g_j]$, then we conjecture that for large $\ell$

$$\log \det \left[ I + \prod_{j=1}^{n} T_\ell[g_j] \right] \sim A\ell \,, \tag{36}$$

where the coefficient $A$ is

$$A = \int_0^{2\pi} \frac{dk}{2\pi} \log \det \left[ I + \prod_{j=1}^{n} g_j(k) \right] \,, \tag{37}$$

provided $\det \left[ I + \prod_{j=1}^{n} g_j(k) \right] \neq 0$. We refer the reader to Appendix E for a discussion on the intuition behind this result.

The second relevant property for our computations concerns the inverse matrix $T_\ell[g]^{-1}$. In general, the inverse of a block Toeplitz matrix is not a block Toeplitz matrix. However, we have checked numerically the following result, which can be derived as a corollary of the conjecture (37) as shown in Appendix E. If we further include in the product of matrices $T_\ell[g_j]$ the inverse $T_\ell[g_j']^{-1}$ of other block Toeplitz matrices with invertible symbol $g_j'(k)$, i.e. $\det[g_j'(k)] \neq 0$ for all $j$, then we conjecture that

$$\log \det \left[ I + \prod_{j=1}^{n} T_\ell[g_j] T_\ell[g_j']^{-1} \right] \sim A'\ell \,, \tag{38}$$

where $A'$ can be calculated from

$$A' = \int_0^{2\pi} \frac{dk}{2\pi} \log \det \left[ I + \prod_{j=1}^{n} g_j(k) g_j'(k)^{-1} \right] \,. \tag{39}$$

We stress that this result holds only in the limit $\ell \to \infty$ and we have tested its validity numerically for arbitrary choices of the symbols $g_j(k), g_j'(k)$.

To derive the time evolution after the quench of the charged moments $Z_n(\boldsymbol{\alpha}, t)$ and, therefore, of the entanglement asymmetry $\Delta S_A^{(n)}(t)$, we apply the following strategy. From the determinant of Eq. (14), we can analytically deduce the asymptotic behaviour for large $\ell$ of $Z_n(\boldsymbol{\alpha}, t)$ at $t = 0$ and $t \to \infty$ using the properties (36) and (38) described above. With these results and applying the quasi-particle picture, we then obtain the exact analytic expression of $Z_n(\boldsymbol{\alpha}, t)$ in the scaling limit $t, \ell \to \infty$ with $\zeta = t/\ell$ fixed. In what follows, we discuss in detail the case $n = 2$, and then we generalise the results to any integer $n \geq 2$.

### 3.3 Calculation of the time evolution

For $n = 2$, Eq. (14) simplifies to

$$Z_2(\alpha, t) = \sqrt{\det \left( \frac{I + \Gamma_\alpha(t)\Gamma_{-\alpha}(t)}{2} \right)} \,, \tag{40}$$

where $\Gamma_\alpha(t) = \Gamma(t)e^{i\alpha n_A}$ and $\alpha \equiv \alpha_1 - \alpha_2$. If the $U(1)$ symmetry is broken, then $\Gamma_\alpha(t)$ and $\Gamma_{-\alpha}(t)$ do not commute and, therefore, $Z_2(\alpha, t) \neq Z_2(0, t)$. While the matrix $\Gamma_\alpha(t)$ is block Toeplitz with symbol $\mathcal{G}_{\alpha,t}(k, \theta) = \mathcal{G}_t(k, \theta)e^{i\alpha(\sigma_z \otimes I)}$, the same is not true for the product $\Gamma_\alpha(t)\Gamma_{-\alpha}(t)$. Therefore, we have to apply the conjecture of Eq. (36) to determine $Z_2(\alpha, t)$ before the quench and its stationary value when $t \to \infty$.

At $t = 0$, if we employ the conjecture (36) in Eq. (40), we have

$$\log Z_2(\alpha, t = 0) \sim \frac{\ell}{4} \int_0^{2\pi} \frac{dk}{2\pi} \log \det \left[ \frac{I + \mathcal{G}_{\alpha,0}(k,\theta)\mathcal{G}_{-\alpha,0}(k,\theta)}{2} \right]. \tag{41}$$

Inserting the explicit expression of the symbol $\mathcal{G}_{\alpha,0}(k,\theta)$, which is given in Eq. (33), we directly find

$$\log Z_2(\alpha, t = 0) \sim \frac{\ell}{2} \int_0^{2\pi} \frac{dk}{2\pi} \log \left[ 1 - \sin^2 \alpha \left( f_{11}(k,\theta)^2 + f_{12}(k,\theta)^2 \right) \right]. \tag{42}$$

On the other hand, to obtain the stationary value of $Z_2(\alpha, t)$ at large times, we can average the time dependent terms in the symbol $\mathcal{G}_t(k,\theta)$ of $\Gamma(t)$, as we did in Eq. (34). Thus employing again the conjecture of Eq. (36), we have

$$\log Z_2(\alpha, t \to \infty) \sim \frac{\ell}{4} \int_0^{2\pi} \frac{dk}{2\pi} \log \det \left[ \frac{I + \mathcal{G}_{\alpha,t\to\infty}(k,\theta)\mathcal{G}_{-\alpha,t\to\infty}(k,\theta)}{2} \right]. \tag{43}$$

Using the time-averaged symbol of Eq. (34), we finally get the stationary behaviour of the charged moments $Z_2(\alpha, t)$ at large times after the quench,

$$\log Z_2(\alpha, t \to \infty) \sim \frac{\ell}{2} \int_0^{2\pi} \frac{dk}{2\pi} \log \left[ h_2(n_+(k,\theta)) h_2(n_-(k,\theta)) - f_{11}(k,\theta)^2 \sin^2 \alpha \right], \tag{44}$$

where $n_\pm(k,\theta) = (g_{12}(k,\theta) \pm f_{11}(k,\theta) + 1)/2$, note that $n_-(k,\theta) = n_+(-k,\theta)$, and

$$h_n(x) = x^n + (1-x)^n . \tag{45}$$

For integer $n > 2$, we cannot remove the inverse matrix $(I - \Gamma(t))^{-1}$ in Eq. (14), as happened for $n = 2$, c.f. Eq. (40). Therefore, one may resort to the conjecture of Eq. (38) to derive the asympotic behaviour of $Z_n(\boldsymbol{\alpha}, t)$ at the initial time and its stationary value after the quench. Unfortunately, the symbol $I - \mathcal{G}_t(k,\theta)$ of the matrix $I - \Gamma(t)$ at $t = 0$ is not invertible and we cannot apply Eq. (38) in that case. Nevertheless, observe that the charged moments $Z_n(\boldsymbol{\alpha}, t = 0)$ of the tilted ferromagnet factorise in the Rényi replica index according to Eqs. (16) and (17). We conjecture that the charged moments of the tilted Néel state admit a similar decomposition that we have numerically checked; that is,

$$\log Z_n(\boldsymbol{\alpha}, t = 0) \sim \frac{\ell}{2} \int_{-\pi}^{\pi} \frac{dk}{2\pi} \log \prod_{j=1}^{n} f_k^{\mathrm{N}}(\theta, \alpha_{j,j+1}). \tag{46}$$

The function $f_k^{\mathrm{N}}(\theta, \alpha)$ can be straightforwardly deduced from the case $n = 2$ analysed before, see Eq. (42),

$$f_k^{\mathrm{N}}(\theta, \alpha) = 1 + m_{\mathrm{N}}(k,\theta) \sin(\alpha), \tag{47}$$

where

$$m_{\mathrm{N}}(k,\theta) = \sqrt{f_{11}(k,\theta)^2 + f_{12}(k,\theta)^2} = \frac{\sin(k/2)\sin^2(\theta)\sqrt{4\cos^2(k/2)\cos^2\theta + \sin^4\theta}}{1 + 2\cos(k)\cos^2\theta + \cos^4\theta}. \tag{48}$$

On the other hand, to calculate the stationary value of $Z_n(\boldsymbol{\alpha}, t)$ at $t \to \infty$, we can take the time average of the correlation $\Gamma(t)$, whose symbol $\mathcal{G}_{t\to\infty}(k)$ was obtained in Eq. (34). It turns out that the time-averaged symbol $I - \mathcal{G}_{t\to\infty}(k,\theta)$ of the matrix $I - \Gamma(t)$ — whose inverse enters

in the determinant formula (14) for $Z_n(\boldsymbol{\alpha}, t)$— is invertible. This means that, in the large time limit, we can apply the conjecture (38) to Eq. (14),

$$\log Z_n(\boldsymbol{\alpha}, t \to \infty) \sim \frac{\ell}{4} \int_0^{2\pi} \frac{dk}{2\pi} \log \det \left[ \left( \frac{I - \mathcal{G}_{t \to \infty}(k)}{2} \right)^n \left( I + \prod_{j=1}^n \mathcal{W}_j(k) \right) \right], \quad (49)$$

where $\mathcal{W}_j(k)$ is the $4 \times 4$ symbol $\mathcal{W}_j(k) = (I + \mathcal{G}_{t \to \infty}(k))(I - \mathcal{G}_{t \to \infty}(k))^{-1} e^{i\alpha_{j,j+1}(\sigma_z \otimes I)}$. Plugging the explicit expression (34) of $\mathcal{G}_{t \to \infty}(k)$ and calculating directly the determinant, we find

$$\log Z_n(\boldsymbol{\alpha}, t \to \infty) \sim \frac{\ell}{2} \int_0^{2\pi} \frac{dk}{2\pi} \log \left[ h_n(n_+(k, \theta)) h_n(n_-(k, \theta)) - f_{11}(k, \theta)^2 \tilde{h}_n(\boldsymbol{\alpha}, k, \theta) \right], \quad (50)$$

where

$$\tilde{h}_n(\boldsymbol{\alpha}, k, \theta) = \sum_{j=1}^{\frac{n - \mathrm{mod}(n,2)}{2}} \frac{f_{11}(k, \theta)^{2j-2} (n_+(k, \theta) + n_-(k, \theta) - 2n_+(k, \theta) n_-(k, \theta))^{n-2j}}{2^{n-2}}$$

$$\times \sum_{1 \le p_1 < p_2 < \cdots < p_{2j} \le n} \sin^2\left( \alpha_{p_1} - \alpha_{p_2} + \cdots - \alpha_{p_{2j}} \right). \quad (51)$$

This result agrees with the one obtained for the case $n = 2$ in Eq. (44). Moreover, when $\boldsymbol{\alpha} = \mathbf{0}$, we must recover the stationary value of the Rényi entanglement entropies. For free systems, it is expected that, at large times after the quench, the Rényi entanglement entropy behaves as [38]

$$S^{(n)}(\rho_A(t \to \infty)) = \frac{1}{1-n} \log Z_n(\mathbf{0}, t \to \infty) \sim \frac{\ell}{1-n} \int_0^{2\pi} \frac{dk}{2\pi} \log[h_n(n(k))], \quad (52)$$

where $n(k)$ is the density of occupied modes in the post-quench stationary state. In fact, it is easy to check that Eq. (50) leads to Eq. (52) with $n(k) = n_+(k, \theta)$, see also Sec. 5.

Once we have the expression of the charged moments $Z_n(\boldsymbol{\alpha}, t)$ at the initial time and its stationary behaviour at large times, Eqs. (46) and (50), we can exploit the quasi-particle picture of entanglement to reconstruct its full time evolution [39–41]. According to it, the quench creates quasi-particle excitations, in particular pairs of entangled quasi-particles emitted from the same point that propagate ballistically in opposite directions with momentum $\pm k$. Therefore, the entanglement generated after the quench is proportional to the pairs of entangled quasi-particles produced in the quench that are shared by the subsystem $A$ and its complementary $B$. Then the integrand of Eq. (52) is the contribution to the Rényi entropy of a pair of entangled excitations with momentum $k$. Since the quasi-particles propagate at a finite velocity $v_k = |\epsilon'_{k/2}|$, the number of entangled pairs of excitations with momentum $k$ shared by $A$ and $B$ at time $t$ is given by $\min(2tv_k, \ell)$. We remark that the initial state $|\Psi(0)\rangle$ is invariant under two-site translations and, when the two-point correlation $\Gamma(t)$ is cast as a block Toeplitz matrix, the Brillouin zone of the fully translationally invariant post-quench Hamiltonian (9) is halved. For this reason, the quasi-particles propagate with velocity $|\epsilon'_{k/2}|$ rather than $|\epsilon'_k|$, as it occurs for the quench from the tilted ferromagnetic state in Eq. (17). Then one finds [38]

$$S^{(n)}(\rho_A(t)) \sim \frac{\ell}{1-n} \int_0^{2\pi} \frac{dk}{2\pi} \min(2\zeta v_k, 1) \log[h_n(n_+(k, \theta))]. \quad (53)$$

The same idea can be applied to deduce the time evolution of the charged moments $Z_n(\boldsymbol{\alpha}, t)$. In this case, we have that $\log Z_n(\boldsymbol{\alpha}, t)$ does not vanish at $t = 0$. Therefore, if we consider

the difference between $\log Z_n(\boldsymbol{\alpha}, t)$ at $t = 0$ and $t \to \infty$, obtained in Eqs. (46) and (50) respectively,

$$\log\left(\frac{Z_n(\boldsymbol{\alpha}, t\to\infty)}{Z_n(\boldsymbol{\alpha}, t=0)}\right) \sim \frac{\ell}{2}\int_0^{2\pi}\frac{dk}{2\pi}\log\left[\frac{h_n(n_+(k,\theta))h_n(n_-(k,\theta))-f_{11}(k,\theta)^2\tilde{h}_n(\boldsymbol{\alpha},k,\theta)}{\prod_{j=1}^n(1+m_N(k,\theta)\sin\alpha_{j,j+1})}\right], \quad (54)$$

then, in the light of the quasi-particle picture, the integrand of this expression can be interpreted as the contribution of an entangled pair of excitations with momentum $\pm k$ created in the quench. Counting the number of such pairs shared between $A$ and $B$ at time $t$, as we have done for the Rényi entanglement entropy, one can conclude that

$$\log\left(\frac{Z_n(\boldsymbol{\alpha}, t)}{Z_n(\boldsymbol{\alpha}, t=0)}\right) \sim$$
$$\frac{\ell}{2}\int_0^{2\pi}\frac{dk}{2\pi}\min(2\zeta v_k, 1)\log\left[\frac{h_n(n_+(k,\theta))h_n(n_-(k,\theta))-f_{11}(k,\theta)^2\tilde{h}_n(\boldsymbol{\alpha},k,\theta)}{\prod_{j=1}^n(1+m_N(k,\theta)\sin\alpha_{j,j+1})}\right]. \quad (55)$$

Observe that, when we take $\boldsymbol{\alpha} = \mathbf{0}$, this result agrees with the time evolution of the Rényi entanglement entropy reported in Eq. (53).

In conclusion, we obtain that, in the scaling limit $t, \ell \to \infty$ with $\zeta = t/\ell$ finite, the exact time evolution of the charged moments $Z_n(\boldsymbol{\alpha}, t)$ after the quench from the tilted Néel state is

$$Z_n(\boldsymbol{\alpha}, t) = Z_n(\mathbf{0}, t)e^{\ell(A_n(\boldsymbol{\alpha})+B_n(\boldsymbol{\alpha},\zeta)+B'_n(\boldsymbol{\alpha},\zeta))}, \quad (56)$$

where

$$A_n(\boldsymbol{\alpha}) = \int_0^{2\pi}\frac{dk}{4\pi}\log\prod_{j=1}^n\left[1+m_N(k,\theta)\sin(\alpha_{j,j+1})\right], \quad (57)$$

$$B_n(\boldsymbol{\alpha},\zeta) = -\int_0^{2\pi}\frac{dk}{4\pi}\min(2\zeta v_k, 1)\log\prod_{j=1}^n\left[1+m_N(k,\theta)\sin(\alpha_{j,j+1})\right], \quad (58)$$

and

$$B'_n(\boldsymbol{\alpha},\zeta) = \int_0^{2\pi}\frac{dk}{4\pi}\min(2\zeta v_k, 1)\log\left[1-\frac{f_{11}(k,\theta)^2\tilde{h}_n(\boldsymbol{\alpha},k,\theta)}{h_n(n_+(k,\theta))h_n(n_-(k,\theta))}\right]. \quad (59)$$

It is interesting to compare the result of Eq. (56) with the one of Eq. (16) for a chain initially prepared in the tilted ferromagnetic state. The terms $A_n(\boldsymbol{\alpha})$ and $B_n(\boldsymbol{\alpha}, \zeta)$ display the same behaviour as in the tilted ferromagnet and $B_n(\boldsymbol{\alpha}, \zeta) \to -A_n(\boldsymbol{\alpha})$ when $\zeta \to \infty$. However, the most remarkable difference is the appearance of the extra term $B'_n(\boldsymbol{\alpha}, \zeta)$, which in general does not vanish in the limit $\zeta \to \infty$. Therefore, for the tilted Néel state, $Z_n(\boldsymbol{\alpha}, t)$ does not tend to the neutral moments $Z_n(\mathbf{0}, t)$ at large times; except if $\theta = \pi/2$, for which $f_{11}(k, \pi/2) = 0$ and the term $B'_n(\boldsymbol{\alpha}, \zeta)$ cancels for any $\zeta$. As we discuss in the following sections, this fact implies that the entanglement asymmetry $\Delta S_A^{(n)}(t)$ does not cancel when $t \to \infty$, which indicates that the $U(1)$ symmetry is not restored if $\theta \neq \pi/2$.

In Fig. 2, we check numerically the time evolution of the charged moments predicted by Eq. (56) in the cases $n = 2$ and $n = 3$. The points are the exact numerical values of $Z_n(\boldsymbol{\alpha}, t)$ computed using the determinant formula of Eq. (14), while the solid curves correspond to Eq. (56). We obtain an excellent agreement. For $n = 3$, $\theta = 4/5$ and $\ell = 100$, the numerical point around $t/\ell = 0.5$ deviates from the analytical prediction due to numerical errors originated in the calculation of the inverse matrix $(I - \Gamma(t))^{-1}$ that appears in Eq. (14). These errors become relevant for large $\ell$ in the regions where $Z_n(\boldsymbol{\alpha}, t)$ presents a peak like in this case.

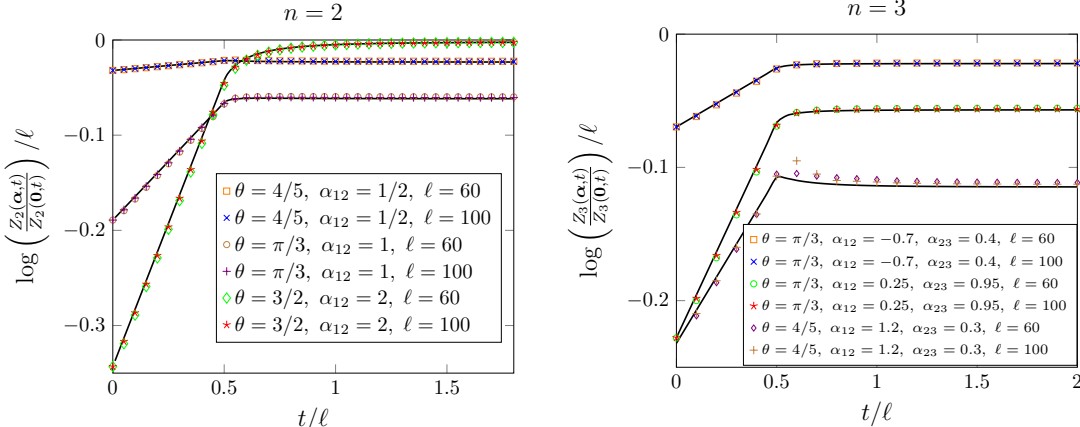

Figure 2: Time evolution of the charged moments $Z_n(\boldsymbol{\alpha}, t)$ as a function of $t/\ell$ for $n = 2$ (left panel) and $n = 3$ (right panel) after the quench from the tilted Néel state. The curves represent the quasi-particle prediction of Eq. (56) for several tilting angles $\theta$ and $\alpha_{j,j+1}$. The points are the exact numerical values of the charged moments obtained using Eq. (14) and taking different subsystem lengths $\ell$.

# 4 Entanglement asymmetry after the quench from the tilted Néel state

In this section, we employ the results for the charged moments previously obtained to analyse the time evolution of the Rényi entanglement asymmetry (4) after the quench (8) from the tilted Néel state.

Combining Eqs. (4) and (11), the entanglement asymmetry $\Delta S_A^{(n)}(t)$ can be derived from the Fourier transform of the charged moments $Z_n(\boldsymbol{\alpha}, t)$. In Fig. 3, the solid curves represent the resulting entanglement asymmetry for different tilting angles $\theta$ when we insert the analytic expression (56) for $Z_n(\boldsymbol{\alpha}, t)$ in Eq. (11), which is exact for any $\zeta = t/\ell$ in the limit $t, \ell \to \infty$. The symbols in the plots correspond to the exact numerical values of the asymmetry, showing a very good agreement with the quasi-particle prediction.

As clear from Fig. 3, the most remarkable feature of the dynamics of the entanglement asymmetry after the quench is that it does not vanish when $t \to \infty$, but it saturates to a value that depends on the tilting angle $\theta$. The exception is the case $\theta = \pi/2$, in which $\Delta S_A^{(n)}(t)$ does tend to zero at large times. In other words, the $U(1)$ symmetry associated to the transverse magnetisation $Q$ is not restored after the quench, unless the symmetry is initially maximally broken, i.e. when $\theta = \pi/2$.

The initial and the stationary value of $\Delta S_A^{(n)}(t)$ for large subsystem sizes $\ell$ can be determined from the analytic expression (56) of the charged moments as follows. In these two cases, once we plug Eq. (56) into Eq. (11), we can solve in the large $\ell$ limit the multi-dimensional integral using the saddle point approximation. In principle, this would require to find the solutions $\boldsymbol{\alpha}^*$ of

$$\nabla_{\boldsymbol{\alpha}}[A_n(\boldsymbol{\alpha}) + B_n(\boldsymbol{\alpha}, \zeta) + B_n'(\boldsymbol{\alpha}, \zeta)] = 0, \tag{60}$$

at $\zeta = 0$ and $\zeta \to \infty$ respectively, and verify that they are maxima of $Z_n(\boldsymbol{\alpha}, t)$ in each case. Since the solution to this equation is not an easy goal to pursue, in order to fix the ideas, we focus on the case $n = 2$.

At $t = 0$, we can employ the expression for the charged moments obtained in Eq. (46). For $n = 2$, if $\alpha \in [-\pi, \pi]$, the function $Z_2(\alpha, 0)$ has saddle points at $\alpha^* = 0, \pi$. Performing a series

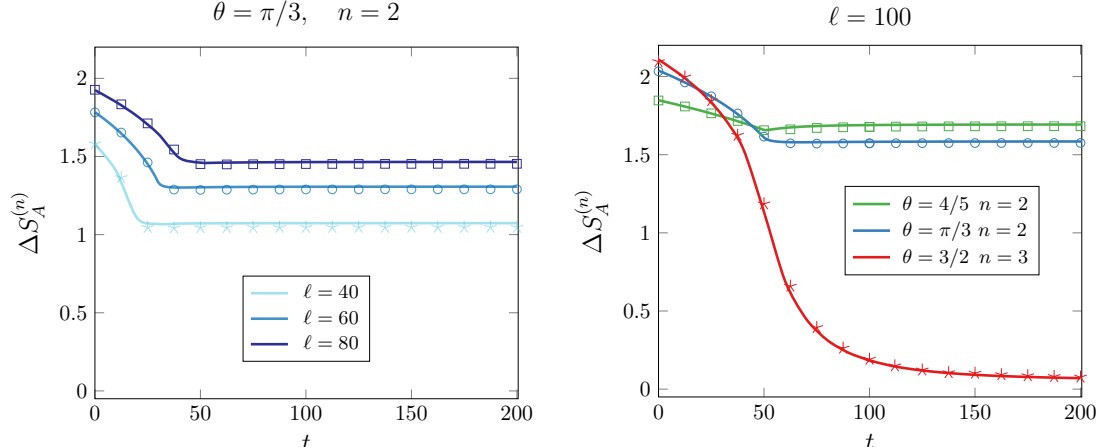

Figure 3: Time evolution of the Rényi entanglement asymmetry after the quench (8) from the tilted Néel state. In the left panel, we fix the initial tilting angle to $\theta = \pi/3$ and we consider several subsystem sizes. In the right panel, we take different tilting angles $\theta$ and Rényi index $n$ for the same subsystem size $\ell = 100$. The curves have been obtained by plugging in Eq. (4) the Fourier transform (11) of our quasi-particle prediction (56) for $Z_n(\boldsymbol{\alpha}, t)$. The symbols are the exact numerical values.

expansion around them up to quadratic order, we obtain

$$\log Z_2(\alpha, t = 0) = \log Z_2(\alpha^*, t = 0) - \frac{(\alpha - \alpha^*)^2}{2} \ell g_0(\theta) + O((\alpha - \alpha^*)^4), \tag{61}$$

where

$$g_0(\theta) = \int_0^{2\pi} \frac{dk}{2\pi} (f_{11}(k, \theta)^2 + f_{12}(k, \theta)^2). \tag{62}$$

By doing the Fourier transform (11) of the quadratic approximations of Eq. (61) around each saddle point, we get

$$\mathrm{Tr}(\rho_{A,Q}^2(t = 0)) = \frac{\mathrm{Tr}(\rho_A^2(t = 0))}{\sqrt{\pi \ell g_0(\theta)/2}} + O(\ell^{-3/2}), \tag{63}$$

and, plugging it into Eq. (4), we obtain that the $n = 2$ entanglement asymmetry behaves at time $t = 0$ as

$$\Delta S_A^{(2)}(t = 0) = \frac{1}{2} \log \ell + \frac{1}{2} \log \frac{\pi g_0(\theta)}{2} + O(\ell^{-1}). \tag{64}$$

For larger integer values of $n$, one can follow a similar reasoning. In this case, Eq. (11) is in principle a $n$-dimensional integral. Applying the neutrality condition $\sum_{j=1}^{n} \alpha_{jj+1} = 0$, it reduces to a $(n-1)$-fold integral and, by doing the change of variables $\beta_j = \alpha_{jj+1}$, we obtain

$$\mathrm{Tr}\rho_{A,Q}^n = \int_{-\pi}^{\pi} \frac{d\beta_1 \cdots d\beta_{n-1}}{(2\pi)^{n-1}} \mathrm{Tr}(\rho_A e^{i\beta_1 Q_A} \rho_A e^{i\beta_2 Q_A} \cdots \rho_A e^{-i(\sum_{i=1}^{n-1} \beta_i) Q_A}). \tag{65}$$

Using the results in Eq. (46), in the large $\ell$ limit, we get

$$\frac{\mathrm{Tr}(\rho_{A,Q}^n(t = 0))}{\mathrm{Tr}(\rho_A^n(t = 0))} \sim \int_{-\pi}^{\pi} \frac{d\beta_1 \cdots d\beta_{n-1}}{(2\pi)^{n-1}} e^{\ell \left[ \sum_{j=1}^{n-1} A_1(\beta_j) + A_1(-\sum_{j=1}^{n-1} \beta_j) \right]}. \tag{66}$$

In order to evaluate this $(n-1)$-fold integral, we should take into account that the number of saddle points in the region of integration is $2^{n-1}$ and that, at leading order, each one gives the

same contribution. Therefore, if we expand the exponent of the integrand around them until quadratic order, we have

$$\frac{\text{Tr}(\rho_{A,Q}^n(t=0))}{\text{Tr}(\rho_A^n(t=0))} \sim 2^{n-1} \int_{-\infty}^{\infty} \frac{d\beta_1 \cdots d\beta_{n-1}}{(2\pi)^{n-1}} e^{-\frac{\ell g_0(\theta)}{2}\left(\sum_{j=1}^{n-1}\beta_j^2 + \sum_{j<j'}\beta_j\beta_{j'}\right)}. \tag{67}$$

Finally, applying the properties of Gaussian integrals,

$$\frac{\text{Tr}(\rho_{A,Q}^n(t=0))}{\text{Tr}(\rho_A^n(t=0))} = \frac{2^{n-1}}{\sqrt{\pi\ell g_0(\theta)}^{n-1}\sqrt{n}} + O(\ell^{-(n+1)/2}), \tag{68}$$

and we then find

$$\Delta S_A^{(n)}(t=0) = \frac{1}{2}\log\ell + \frac{1}{2}\log\frac{\pi n^{1/(n-1)}g_0(\theta)}{4} + O(\ell^{-1}). \tag{69}$$

We remark that this result holds for the entanglement asymmetry of the cat state $|\Psi(0)\rangle$ in Eq. (6), while the expression is slightly different if we consider the non-cat state $|N,\theta\rangle$ in Eq. (7). Moreover, the integral (62) that gives the term $g_0(\theta)$ can be explicitly computed. By performing the change of variables $z = e^{ik}$, it can be re-expressed as a contour integral in the complex plane $z$ and, applying the residue theorem, we eventually obtain that

$$g_0(\theta) = \frac{\sin^2(\theta)}{2}. \tag{70}$$

From this equality, we find that, at time $t = 0$, the entanglement asymmetry for the cat tilted Néel, Eq. (69), and the ferromagnetic state, Eq. (20), are the same in the large interval limit. It is interesting to note that the term $g_0(\theta)$, as an integral in momentum space, is given by the square of the eigenvalues of the symbol $\mathcal{F}_t(k,\theta)$ at $t = 0$ that generates the correlations $\langle\Psi(0)|c_j^\dagger c_{j'}^\dagger|\Psi(0)\rangle$, see Eq. (30).

In the large $t$ limit, we can repeat the same steps. Now we can use the stationary value of the charged moments provided by Eq. (50). If $n = 2$, this function presents two saddle points, at $\alpha^* = 0, \pi$, whose leading contributions in $\alpha$ read

$$\log Z_2(\alpha, t \to \infty) = \log Z_2(\alpha^*, t \to \infty) - \frac{(\alpha-\alpha^*)^2}{2}\ell g_\infty^{(2)}(\theta) + O((\alpha-\alpha^*)^4), \tag{71}$$

where

$$g_\infty^{(2)}(\theta) = \int_0^{2\pi} \frac{dk}{2\pi} \frac{f_{11}(k,\theta)^2}{h_2(n_+(k,\theta))h_2(n_-(k,\theta))}. \tag{72}$$

Notice that this expansion is analogous to the one of Eq. (61) for $t = 0$. Then we can directly conclude that the stationary $n = 2$ entanglement asymmetry after the quench has the form

$$\Delta S_A^{(2)}(t \to \infty) = \frac{1}{2}\log\ell + \frac{1}{2}\log\frac{\pi g_\infty^{(2)}(\theta)}{2} + O(\ell^{-1}). \tag{73}$$

By repeating similar steps, we can also obtain an analytical prediction for the stationary value of the Rényi entanglement asymmetry for larger integer $n$, which is given by

$$\Delta S_A^{(n)}(t \to \infty) = \frac{1}{2}\log\ell + \frac{1}{2}\log\frac{\pi n^{1/(n-1)}g_\infty^{(n)}(\theta)}{4} + O(\ell^{-1}). \tag{74}$$

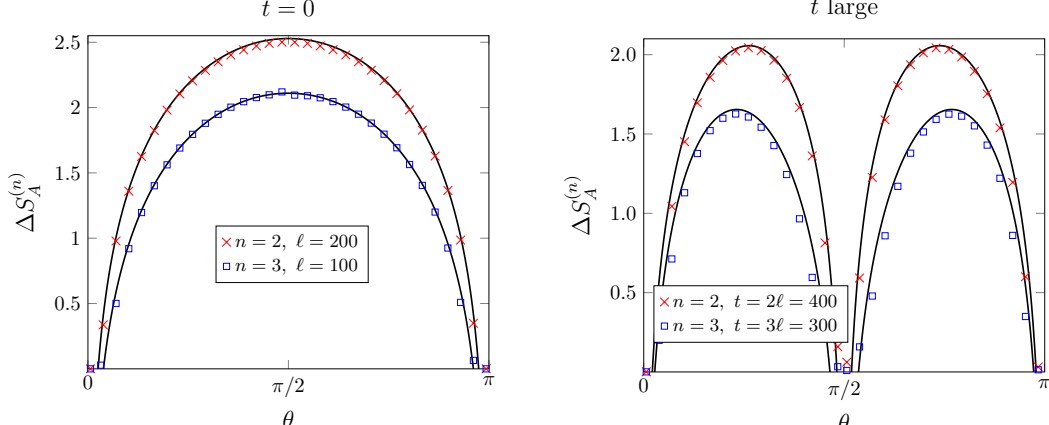

Figure 4: Rényi entanglement asymmetry as a function of the tilting angle $\theta \in [0, \pi]$ at initial time (left) and at a fixed large time after the quench (right). We consider different Rényi index $n$ and subsystem lengths $\ell$. The black curves in the left panel correspond to the asymptotic expression of the entanglement asymmetry at $t = 0$ obtained in Eq. (69) while in the right panel they are the prediction (74) for its stationary value with $g_\infty^{(n)}$ given by Eq. (72) for $n = 2$ and by Eq. (75) for $n = 3$. The symbols are the exact numerical values.

As at initial time, the stationary entanglement asymmetry grows logarithmically with the subsystem length $\ell$. The $\ell$-independent term is instead different and, moreover, it shows a nontrivial dependence on the Rényi index $n$; for example, for $n = 3$,

$$g_\infty^{(3)}(\theta) = \int_0^{2\pi} \frac{dk}{2\pi} \frac{f_{11}(k,\theta)^2 (n_+(k,\theta) + n_-(k,\theta) - 2n_+(k,\theta)n_-(k,\theta))}{h_3(n_+(k,\theta))h_3(n_-(k,\theta))} . \tag{75}$$

Although for any integer $n$ we can reconstruct an expression for $g_\infty^{(n)}$, it gets more and more cumbersome as $n$ increases and a closed analytic form cannot be obtained.

In Fig. 4, we report the profile of the entanglement asymmetry for $n = 2, 3$ in terms of the tilting angle $\theta$ both before (left panel) and at large times after the quench (right panel). The symbols indicate the exact numerical value and the solid curves represent the expressions for large $\ell$ found in Eq. (69) for $t = 0$ (left panel) and in Eq. (74) for the stationary regime (right panel). The agreement between the curves and the numerical points worsens as $\Delta S_A^{(n)}$ tends to zero, i.e. when the symmetry is restored. As we have already discussed, this happens for $\theta = 0, \pi$ at $t = 0$ and for $\theta = 0, \pi/2, \pi$ when $t \to \infty$. This can be understood also from our analytical prediction in Eqs. (69) and (74). The functions $g_0(\theta)$ and $g_\infty^{(n)}(\theta)$ vanish at $\theta = 0, \pi$ and $\theta = 0, \pi/2, \pi$ respectively and Eqs. (69), (74) are not well defined when the symmetry is respected. In fact, the limits $\ell \to \infty$ and $\theta \to 0, \pi/2, \pi$ do not commute: to properly get $\Delta S_A^{(n)} = 0$ when the symmetry is recovered, one should fix first the value of $\theta$ in evaluating the correlators (23), (34) and then consider the large $\ell$ regime in Eq. (14). We observe that, when $t = 0$, $\Delta S_A^{(n)}$ is a monotonic function of $\theta$ between $\theta = 0$ and $\theta = \pi/2$. Notice that, according to the asymptotic expressions (69) and (74) of $\Delta S_A^{(n)}$ at $t = 0$ and $t = \infty$ plotted in Fig. 4, for a given tilting angle $\theta$ the initial entanglement asymmetry is not always larger than its stationary value after the quench. That is, the entanglement asymmetry does not always decrease after the quench but it can also increase depending on $\theta$. Therefore, a phenomenon similar to the quantum Mpemba effect found when the initial state is the tilted ferromagnetic configuration (15) cannot be in general observed when the system is prepared in the tilted

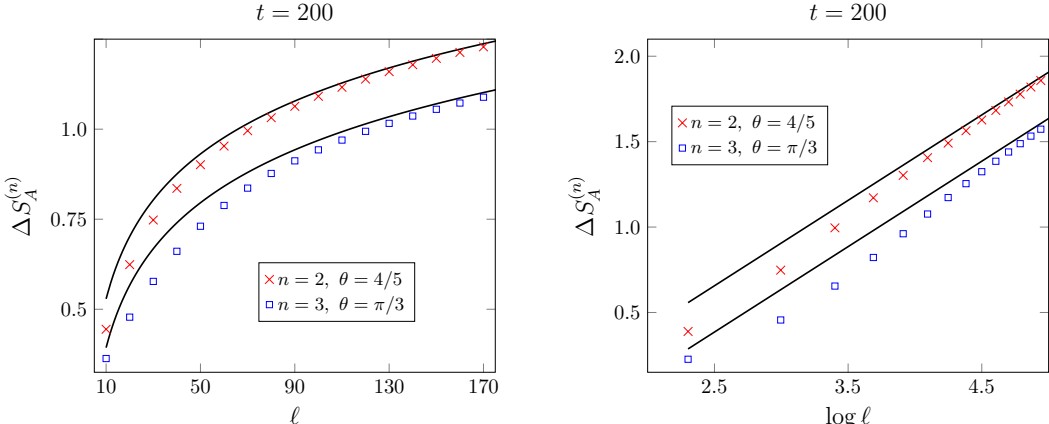

Figure 5: Rényi entanglement asymmetry at a large time after the quench (8) from the tilted Néel state as a function of the subsystem size $\ell$ (left panel) and $\log \ell$ (right panel) for different Rényi index $n$ and initial tilting angle $\theta$. The black curves correspond to the prediction (74) for its stationary value with $g_\infty^{(n)}$ given by Eq. (72) for $n = 2$ and by Eq. (75) for $n = 3$ while the symbols are the exact numerical values.

Néel state and quenched to the XX spin chain Hamiltonian.

In order to verify the logarithmic behaviour in $\ell$ of Eq. (74), we compare it with the exact numerical results in Fig. 5 by varying the interval length. As the subsystem size increases, we observe that the agreement between the numerics and our theoretical predictions improves, despite we are using finite values both for $t$ and $\ell$ and our predictions are valid in the scaling limit $t, \ell \to \infty$ with $t/\ell$ finite. In particular, we choose $t = 200$ because, from the right panel of Fig. 3, it corresponds to the saturation regime of the asymmetry. However, we remind the reader that our results hold in the limit $\ell \to \infty$, so a good agreement is visible only for large values of $\ell$.

## 5 Description in terms of the post-quench stationary state

The quasi-particle picture employed in the previous sections makes use of the density $n(k)$ of occupied modes in the post-quench stationary state. In free or integrable models, the latter can be obtained by a description of the stationary state in terms of a Generalised Gibbs Ensemble (GGE), taking account all of the conserved local or quasilocal charges. As we will see now, the situation here is particularly subtle, due to the presence of a non-Abelian set of conserved charges for the Hamiltonian (9), a feature also known as *superintegrability* [42]. Non-Abelian charges also found application in other contexts such as quantum thermodynamics [43–47] and time crystals [48,49].

The charges commuting with the Hamiltonian (9) can be split in four families, which we write in the thermodynamic limit as [50–52]

$$H_m = \frac{1}{2i} \sum_j \left( e^{i\frac{m\pi}{2}} c_j^\dagger c_{j+m} - e^{-i\frac{m\pi}{2}} c_{j+m}^\dagger c_j \right) = \int_0^{2\pi} \frac{dk}{2\pi} \sin\left( m\left( \frac{\pi}{2} - k \right) \right) c_k^\dagger c_k, \quad (76)$$

$$Z_m = \frac{1}{2} \sum_j \left( e^{i\frac{m\pi}{2}} c_j^\dagger c_{j+m} + e^{-i\frac{m\pi}{2}} c_{j+m}^\dagger c_j \right) = \int_0^{2\pi} \frac{dk}{2\pi} \cos\left( m\left( \frac{\pi}{2} - k \right) \right) c_k^\dagger c_k, \quad (77)$$

$$Y_m^\dagger \;=\; \sum_j (-1)^{j+1} c_j^\dagger c_{j+m}^\dagger = \int_0^{2\pi} \frac{dk}{2\pi} e^{ikm} c_k^\dagger c_{\pi-k}^\dagger \,, \tag{78}$$

$$Y_m \;=\; \sum_j (-1)^{j} c_j c_{j+m} = \int_0^{2\pi} \frac{dk}{2\pi} e^{-ikm} c_{\pi-k} c_k \,, \tag{79}$$

(for a system of finite size $L$ with periodic boundary conditions fermion bilinears with $j \leq L < j+m$ come with a different prefactor depending on the global charge $Q$, but we will not need to worry about this here). Note in particular that $H_1$ is (proportional to) the Hamiltonian $H$ and $Z_0$ is the number operator generating the $U(1)$ symmetry.

The charges $H_m$ commute with all others, as is made clear by the Fourier transform to momentum space: since they have a dispersion relation which is odd under $k \to \pi-k$, fermions of momenta $k$ and $\pi-k$ come with opposite $H_m$-eigenvalue, and can therefore be created or destroyed in pairs without affecting the $H_m$ charges [52]. In contrast, the charges $Z_m$, $Y_m$ and $Y_m^\dagger$ form three commuting families, but do not commute with one another (we note in passing that the same structure of non-commuting conserved charges can be found for interacting models, namely for the XXZ models at the "root-of-unity" points $\Delta = \cos\frac{\pi l}{m}$, $m,l \in \mathbb{Z}^*$ [58–62]; historically, these charges appeared first in the field theory literature [63,64]). Another important point is that the charges $H_m$ and $Z_m$ commute with the $U(1)$ charge $Q = Z_0$, while $Y_m$ and $Y_m^\dagger$ do not. Let us now see how these charges affect the GGE description of the post-quench relaxation.

## 5.1 Abelian case

Let us start by briefly reviewing the case of models with a commuting family of conserved charges $\{H_m\}$, as happens for instance in the XXZ chain with generic interaction parameter $\Delta$. It is now well-established in such cases that the late-time steady state following a quantum quench should be described by a GGE entirely specified by the expectation values of all local or quasilocal conserved charges [53–55]. Consequently, local observables relax at late time to

$$\lim_{t\to\infty}\lim_{L\to\infty} \langle\Psi(0)|e^{-iHt}\mathcal{O}_j e^{iHt}|\Psi(0)\rangle = \mathrm{Tr}\big(\rho_{\mathrm{GGE}}\mathcal{O}_j\big) = \langle\Psi_{\mathrm{GGE}}|\mathcal{O}_j|\Psi_{\mathrm{GGE}}\rangle \,, \tag{80}$$

where $\mathcal{O}_j$ is an operator localise around position $j$, $\rho_{\mathrm{GGE}}$ is a density matrix of the form $\rho_{\mathrm{GGE}} \propto \exp(-\sum_m \beta_m H_m)$ where all the Lagrange multipliers $\beta_m$ are characterised by the expectation values of (quasi)local conserved charges, and the last equality states the equivalence of the latter with a *representative state*, namely an eigenstate of all the conserved charges characterised by their expectation values on the initial state [2,56,57]. In Bethe ansatz-integrable models, the representative state is described by a set of densities associated with the various types of quasi-particles, which reduce in the XX chain/free fermionic case to the mode occupation function

$$n(k) = \langle\Psi(0)|c_k^\dagger c_k|\Psi(0)\rangle \,. \tag{81}$$

## 5.2 Non-Abelian case

We now investigate the effect of the additional non-Abelian set of conserved charges in the XX model, namely $\{Z_m\}$, $\{Y_m\}$, $\{Y_m^\dagger\}$, in addition to the usual $\{H_m\}$. We point out that related questions have been addressed in [50,51], where the focus was however on a weak perturbation breaking the non-Abelian symmetry; here instead no perturbation is introduced, and the symmetry is exact.

The non-Abelian symmetry splits the spectrum of the charges $H_m$ into degenerate subspaces, where the charges $Z_m$, $Y_m$ and $Y_m^\dagger$ act non-commutatively [52]. Therefore, specifying

the expectation values of the charges $H_m$ does not uniquely fix a representative state, as can be seen more directly by observing that Bogoliubov rotations of the form

$$
\begin{pmatrix} c_k \\ c^\dagger_{\pi-k} \end{pmatrix} \longrightarrow \begin{pmatrix} \cos r_k & -e^{-i\varphi_k}\sin r_k \\ e^{i\varphi_k}\sin r_k & \cos r_k \end{pmatrix} \begin{pmatrix} c_k \\ c^\dagger_{\pi-k} \end{pmatrix},
$$
$$
r_k, \varphi_k \in [0, 2\pi], \ k \in \left[-\frac{\pi}{2}, \frac{\pi}{2}\right],
\tag{82}
$$

leave the charges $H_m$ invariant, while changing the mode occupation function (81). Alternatively, we can view those transformations as a rotation under the unitary operator

$$
\mathcal{U}_k = \exp\left(r_k(e^{i\varphi_k} c^\dagger_k c^\dagger_{\pi-k} - e^{-i\varphi_k} c_{\pi-k} c_k)\right).
\tag{83}
$$

A complete GGE should therefore include, in addition to the charges $H_m$, a maximal Abelian subset of the remaining charges. Different choices of complete GGEs are conjugated to one another by arbitrary products of rotations of the type (83), and, as we will see now, which choice is to be made crucially depends on the quench protocol under consideration.

**Quench from the tilted ferromagnetic state**  It is instructive to start by revisiting the case of a quench from the tilted ferromagnetic state, recently considered in [5], see also Section 2.2. There it was shown that the $U(1)$ symmetry generated by $Q$ is restored at large time. This can be viewed as a consequence of the fact that the charges $Y_m$ and $Y^\dagger_m$ have zero expectation value, as they are odd under translation while the initial state is translationally invariant.

It is then natural to look for a GGE built out of the maximal set of charges commuting with $Q$, namely, the $\{H_m\}$ and $\{Z_m\}$ charges (this is the same type of GGE as what has been considered in [65], in an interacting setup). As is clear from their expression (76), (77) in terms of the fermionic mode operators, specifying the expectation of those charges amounts to specifying the mode occupation numbers (81), which are found to be [5]

$$
2n(k) - 1 = \frac{2\cos\theta - (1 + (\cos\theta)^2)\cos k}{1 - 2\cos\theta\cos k + (\cos\theta)^2} \equiv \cos\Delta_k.
\tag{84}
$$

As described in Sec. 2.2, combining these densities with the quasi-particle picture (Eqs. (16), (17)) allows us to recover the correct expression for the charged moments of the entanglement asymmetry.

**Quench from the tilted Néel state**  We now turn back to the case of a quench from the tilted Néel state, which is the main focus of this paper. As for the tilted ferromagnet, it is easy to compute the mode occupation numbers by evaluating the various bilinear combinations of fermionic operators in the initial state. The explicit computation has been reported in Appendix D and the final result is

$$
\langle\Psi(0)|c^\dagger_k c_k|\Psi(0)\rangle = \frac{1}{2}\left(1 - \frac{(1 - (\cos\theta)^4)\cos k}{1 + 2(\cos\theta)^2\cos 2k + (\cos\theta)^4}\right).
\tag{85}
$$

However, plugging the resulting mode occupation numbers $n(k)$ into equation (52) for the stationary value of the Rényi entanglement entropy does not recover the result obtained from numerics or from the calculations presented in the previous sections. The reason for this is that in the present case the $U(1)$ symmetry generated by $Q$ is not restored, in other terms the charges $Y^{(\dagger)}_m$ have a non-zero expectation value, as can be seen by computing the off-diagonal conserved charges (again, using the results of the Appendix D)

$$
\langle\Psi(0)|c_{\pi-k}c_k|\Psi(0)\rangle = -\langle\Psi(0)|c^\dagger_k c^\dagger_{\pi-k}|\Psi(0)\rangle = \frac{i}{2}\frac{(\sin\theta)^2\cos\theta\sin 2k}{1 + 2(\cos\theta)^2\cos 2k + (\cos\theta)^4}.
\tag{86}
$$

Let us however make the following observation: applying to all values of $k$ a rotation of the form (82), (83) with $r_k = \frac{\pi}{4}$, $\varphi_k = \frac{\pi}{2}$, namely defining the rotated fermionic operators

$$\widetilde{c}_k = \frac{c_k + i c^\dagger_{\pi-k}}{\sqrt{2}}, \qquad \widetilde{c}^\dagger_{\pi-k} = \frac{c^\dagger_{\pi-k} + i c_k}{\sqrt{2}}, \tag{87}$$

we have

$$\langle\Psi(0)|\widetilde{c}_{\pi-k}\widetilde{c}_k|\Psi(0)\rangle = \frac{1}{2}\langle\Psi(0)|\left(c_{\pi-k}c_k + c^\dagger_k c^\dagger_{\pi-k} + i(c_{\pi-k}c^\dagger_{\pi-k} - c^\dagger_k c_k)\right)|\Psi(0)\rangle = 0, \tag{88}$$

$$\langle\Psi(0)|\widetilde{c}^\dagger_{\pi-k}\widetilde{c}^\dagger_k|\Psi(0)\rangle = \frac{1}{2}\langle\Psi(0)|\left(c^\dagger_{\pi-k}c^\dagger_k + c_k c_{\pi-k} - i(c^\dagger_{\pi-k}c_{\pi-k} - c_k c^\dagger_k)\right)|\Psi(0)\rangle = 0, \tag{89}$$

as resulting from (86) and from the fact that

$$\langle\Psi(0)|c^\dagger_k c_k|\Psi(0)\rangle = \langle\Psi(0)|c_{\pi-k}c^\dagger_{\pi-k}|\Psi(0)\rangle. \tag{90}$$

What Eqs. (88) and (89) mean, is that the rotated $U(1)$ symmetry generated by

$$\widetilde{Q} = \int_0^{2\pi} \frac{dk}{2\pi}\widetilde{c}^\dagger_k\widetilde{c}_k \tag{91}$$

should be restored at late times after the quench. Following the logic discussed above for the tilted ferromagnetic case, we therefore look for a GGE built out a maximal set of charges commuting with $\widetilde{Q}$. Equivalently this can be expressed in terms of the rotated mode occupations $\widetilde{n}(k)$, obtained from:

$$\langle\Psi(0)|\widetilde{c}^\dagger_k\widetilde{c}_k|\Psi(0)\rangle = \langle\Psi(0)|c^\dagger_k c_k|\Psi(0)\rangle - i\langle\Psi(0)|c_{\pi-k}c_k|\Psi(0)\rangle$$
$$= \frac{1}{2}\left(1 - \frac{\sin(\theta)^2\cos(k)}{1+\cos(\theta)^2+2\cos(\theta)\sin k}\right). \tag{92}$$

The corresponding distribution function $\widetilde{n}(k)$ is precisely the function $n_-(2k, \theta)$ which was found in the previous sections to enter the quasi-particle picture. Note that $n_-$ is identified with $\widetilde{n}$ upon replacing in the former $k \mapsto 2k$. The reason is that $\widetilde{n}$ has been calculated using the modes $c_k$, $c^\dagger_k$ that diagonalise the post-quench Hamiltonian $H$ while in the calculation of $n_-$ the Brillouin zone of $H$ is halved, as already explained in Sec. 3.

## 6 Conclusions

In this paper, we considered the quantum quench in the XX spin chain starting from the cat-version of the tilted Néel state given by Eq. (6). This state (both in normal and cat version) explicitly breaks the $U(1)$ symmetry of the XX Hamiltonian. We found that, surprisingly, the $U(1)$ symmetry is not generically restored at large time and this can be traced back to the activation of a non-Abelian set of charges which all break it. We characterised quantitatively the breaking of the symmetry by the recently introduced entanglement asymmetry [5]. By a combination of exact calculations and quasi-particle picture arguments, we have been able to exactly describe the behaviour of the asymmetry at any time after the quench. We obtained that, at large times after the quench, the entanglement asymmetry tends to a non-zero constant value, except at $\theta = \pi/2$, for which it does go to zero. Hence, the $U(1)$ symmetry is not restored unless it is maximally broken at initial time. Finally, we showed that the stationary behaviour is completely captured by a non-Abelian generalised Gibbs ensemble.

We conclude this paper with some outlooks. The first natural question is whether in interacting integrable models there are integrable initial states (in the sense of Ref. [66]) for which symmetries of the post-quench Hamiltonian are not restored. This entirely depends on the structure of the GGE and on the activation of possibly existing non-Abelian charges. For example, it is known that the XXZ spin chain at root of unity (i.e. only in the regime $|\Delta| < 1$) there are non-Abelian charges [59, 62] and they are activated by the tilted Néel. Then, in this case, we expect the non-restoration of the $U(1)$ symmetry. Conversely, in the regime $|\Delta| > 1$ the $U(1)$ symmetry is expected to be always restored. The calculation of the entanglement asymmetry for the XXZ spin-chain should be possible by generalising the approach already used for non-equilibrium symmetry resolved entanglement in these models [16, 17, 67] and work in this direction is already in progress. The situation is instead less clear for other integrable models, such as for example Hubbard or Gaudin-Yang, for which integrable initial states have been recently proposed [68, 69]. Another more difficult question concerns how to describe a (weak) integrability breaking which eventually always leads to symmetry restoration, but with a pre-thermal regime in which the symmetry is broken. It should be possible to study this problem with pre-thermalisation techniques [70–74].

## Acknowledgments

We thank Colin Rylands and Olexei I. Motrunich for useful discussions. We also thank an anonymous referee for useful comments and suggestions which helped us to improve the manuscript.

**Funding information** PC and FA acknowledge support from ERC under Consolidator grant number 771536 (NEMO). SM thanks support from Caltech Institute for Quantum Information and Matter and the Walter Burke Institute for Theoretical Physics at Caltech.

## Appendices

## A Gaussianity of the cat tilted Néel state

In this Appendix, we justify why the reduced density matrix of the cat version $|\Psi(0)\rangle$ of the tilted Néel state in Eq. (7) is a Gaussian state. To prove this, we need to show that $|\Psi(0)\rangle$ satisfies Wick theorem: the $2m$-point fermionic correlation functions decompose into 2-point correlators. Since the eigenstates of a quadratic fermionic Hamiltonian are Slater determinants, they satisfy Wick theorem and their reduced density matrix is Gaussian [31]. It is then enough to verify that $|\Psi(0)\rangle$ is the eigenstate of a quadratic Hamiltonian in terms of the fermionic operators $c_j$, $c_j^\dagger$.

As shown in Refs. [75] and [76], the tilted ferromagnetic state $|F, \theta\rangle$ and its rotation of angle $\pi$ around the $z$-axis $|F, -\theta\rangle$ are degenerate ground states of the XY spin chain,

$$H_{XY} = -\frac{1}{4} \sum_{j=-\infty}^{\infty} \left[ (1+\gamma)\sigma_j^x \sigma_{j+1}^x + (1-\gamma)\sigma_j^y \sigma_{j+1}^y + 2h\sigma_j^z \right], \tag{93}$$

with $\cos^2(\theta) = (1-\gamma)/(1+\gamma)$ and $h^2 + \gamma^2 = 1$. The XY spin chain is quadratic in $c_j$ and $c_j^\dagger$ and it can be diagonalised through certain Bogoliubov fermionic operators $\eta_k$, $\eta_k^\dagger$. Therefore, the eigenstates of $H_{XY}$ are the Slater determinants $|\mathbb{K}\rangle = \prod_{k \in \mathbb{K}} \eta_k^\dagger |0\rangle$ associated to the different

configurations $\mathbb{K}$ of occupied Bogoliubov modes over the vacuum $|0\rangle$, which satisfies $\eta_k |0\rangle = 0$ for all $k$. Since $H_{\mathrm{XY}}$ commutes with the parity operator $P_z = \prod_j \sigma_j^z$, the Slater determinants $|\mathbb{K}\rangle$ are also eigenstates of $P_z$, as shown in Ref. [77]. Therefore, given that $P_z |\mathrm{F}, \theta\rangle = |\mathrm{F}, -\theta\rangle$, the tilted states $|\mathrm{F}, \pm\theta\rangle$ do not correspond to any Slater determinant $|\mathbb{K}\rangle$ but their linear combinations $(|\mathrm{F}, \theta\rangle \pm |\mathrm{F}, -\theta\rangle)/\sqrt{2}$ do.

Now observe that the tilted Néel and ferromagnetic states are related by the linear operator $\widetilde{P}_y = \prod_l (-i\sigma_{2l-1}^y)$ such that $|\mathrm{N}, \theta\rangle = \widetilde{P}_y |\mathrm{F}, \theta\rangle$. Since $\widetilde{P}_y^\dagger \widetilde{P}_y = I$ and $H_{\mathrm{XY}} |\mathrm{F}, \pm\theta\rangle = E_\theta |\mathrm{F}, \pm\theta\rangle$, then the tilted Néel states $|\mathrm{N}, \pm\theta\rangle$ are eigenstates with energy $E_\theta$ of the Hamiltonian $H'_{\mathrm{XY}} = \widetilde{P}_y H_{\mathrm{XY}} \widetilde{P}_y^\dagger$, which has the form [78]

$$H'_{\mathrm{XY}} = -\frac{1}{4} \sum_{j=-\infty}^{\infty} \left[ -(1+\gamma)\sigma_j^x \sigma_{j+1}^x + (1-\gamma)\sigma_j^y \sigma_{j+1}^y - 2h(-1)^j \sigma_j^z \right]. \tag{94}$$

This is the XY spin chain with a staggered transverse magnetic field, which again is quadratic in the fermionic operators $c_j$ and $c_j^\dagger$. Note that $H'_{\mathrm{XY}}$ commutes with $P_z$ and, therefore, the eigenstates described by a Slater determinant must be also eigenstates of $P_z$ [77]. Since $P_z |\mathrm{N}, \pm\theta\rangle = |\mathrm{N}, \mp\theta\rangle$, by the same reasoning as in the tilted ferromagnet, the linear combinations $(|\mathrm{N}, \theta\rangle \pm |\mathrm{N}, -\theta\rangle)/\sqrt{2}$ are eigenstates of $P_z$, thus they can be written as Slater determinants and satisfy Wick theorem.

## B  Details about the derivation of Eq. (14)

In this Appendix, we derive the expression (14) that relates the charged moments $Z_n(\boldsymbol{\alpha})$ defined in Eq. (12) and the two-point correlation function $\Gamma$ introduced in Eq. (13) when the reduced density matrix $\rho_A$ is Gaussian; that is, when

$$\rho_A = \frac{1}{Z} e^{-\frac{1}{2} \sum_{j,j'} c_j^\dagger h_{jj'} c_{j'}}, \quad Z = \mathrm{Tr}\left( e^{-\frac{1}{2} \sum_{j,j'} c_j^\dagger h_{jj'} c_{j'}} \right), \tag{95}$$

where the factor $Z$ ensures that $\mathrm{Tr}(\rho_A) = 1$. In such case, the charged moments $Z_n(\boldsymbol{\alpha})$ are a product of Gaussian operators since the transverse magnetisation is a quadratic operator in terms of $c_j^\dagger$ and $c_j$,

$$Q_A = \frac{1}{2} \sum_{j,j'} c_j^\dagger (n_A)_{jj'} c_{j'}, \tag{96}$$

where $n_A$ is a diagonal matrix with $(n_A)_{2j,2j} = 1$, $(n_A)_{2j-1,2j-1} = -1$,

By the Baker-Campbell-Haussdorf formula, the product of Gaussian operators is Gaussian [33],

$$e^{\frac{1}{2} \sum_{j,j'} c_j^\dagger A_{jj'} c_{j'}} e^{\frac{1}{2} \sum_{j,j'} c_j^\dagger B_{jj'} c_{j'}} = e^{\frac{1}{2} \sum_{j,j'} c_j^\dagger H_{jj'} c_{j'}}, \tag{97}$$

where $H = \log(e^A e^B)$. Moreover, for a general non-Hermitian matrix $H$, we can use that [33]

$$\mathrm{Tr}(e^{\frac{1}{2} \sum_{j,j'} c_j^\dagger H_{jj'} c_j}) = \sqrt{\det(I + e^H)}. \tag{98}$$

The other crucial ingredient is that the single-particle entanglement Hamiltonian $h$ of Eq. (95) can be written in terms of the correlation matrix $\Gamma$ as [31]

$$e^{-h} = \frac{I + \Gamma}{I - \Gamma}. \tag{99}$$

Combining the previous ingedients, we can directly derive Eq. (14). In fact, if we first apply Eqs. (97) and (98) in Eq. (12), we obtain

$$Z_n(\boldsymbol{\alpha}) = Z^{-n} \sqrt{\det\left( I + \prod_{j=1}^{n} e^{-h} e^{i\alpha_{jj+1} n_A} \right)}. \tag{100}$$

Using Eqs. (98) and (99) we have that $Z^{-1} = \sqrt{\det((I - \Gamma)/2)}$ and, therefore,

$$Z_n(\boldsymbol{\alpha}) = \left[\det\left(\frac{I - \Gamma}{2}\right)\right]^{n/2} \left[\det\left(I + \prod_{j=1}^{n} \frac{I + \Gamma}{I - \Gamma} e^{i\alpha_{jj+1}n_A}\right)\right]^{1/2}, \tag{101}$$

which is precisely Eq. (14), upon identifying $\Gamma$ with $\Gamma(t)$ and $W_j(t) = (I + \Gamma(t))(I - \Gamma(t))^{-1}e^{i\alpha_{jj+1}n_A}$.

## C Post-quench two-point correlation functions

In this Appendix, we explicitly derive the two-point correlation matrices of Sec. 3. We first obtain the correlation matrix for the cat tilted Néel state of Eq. (6) and afterwards its post-quench time evolution, Eq. (8). We start by rewriting the tilted Néel state in Eq. (7) as,

$$|\mathrm{N}, \theta\rangle = \left[\left(\cos\frac{\theta}{2}|\downarrow\rangle + \sin\frac{\theta}{2}|\uparrow\rangle\right)\left(\cos\frac{\theta}{2}|\uparrow\rangle - \sin\frac{\theta}{2}|\downarrow\rangle\right)\right]^{\otimes L/2}, \tag{102}$$

where $L$ is the total number of spins of the chain. Using the Jordan-Wigner transformation, we can re-express this state in terms of the fermionic operators $c_j^\dagger$, $c_j$ by identifying $|\uparrow\rangle_j = c_j^\dagger|0\rangle_j$, $|\downarrow\rangle_j = |0\rangle_j$, being $|0\rangle_j$ the fermionic vacuum state. Taking this into account, the calculation of the cat state correlators $\langle\Psi(0)|c_j^\dagger c_{j'}|\Psi(0)\rangle$ and $\langle\Psi(0)|c_j c_{j'}|\Psi(0)\rangle$ is straightfoward. In the thermodynamic limit $L \to \infty$, which is the case of interest for us, the off-diagonal elements $\langle\mathrm{N}, \theta|c_j^\dagger c_{j'}|\mathrm{N}, -\theta\rangle$ and $\langle\mathrm{N}, \theta|c_j c_{j'}|\mathrm{N}, -\theta\rangle$ vanish and, therefore, we can assume

$$\langle\Psi(0)|c_j^\dagger c_{j'}|\Psi(0)\rangle = \langle\mathrm{N}, \theta|c_j^\dagger c_{j'}|\mathrm{N}, \theta\rangle, \tag{103}$$

$$\langle\Psi(0)|c_j c_{j'}|\Psi(0)\rangle = \langle\mathrm{N}, \theta|c_j c_{j'}|\mathrm{N}, \theta\rangle, \tag{104}$$

where we have further applied that the two-point correlators of $|\mathrm{N}, \theta\rangle$ are even functions of $\theta$. Since we we have translational invariance by two-site shifts, it is useful to distinguish between even and odd sites. Let us then introduce the $2 \times 2$ matrices $C_{l,l'}$ and $F_{l,l'}$, with $l, l' = 1, 2, \ldots,$

$$\tilde{C}_{l,l'} = \langle\Psi(0)|\begin{pmatrix} c_{2l-1}^\dagger \\ c_{2l}^\dagger \end{pmatrix}(c_{2l'-1}, c_{2l'})|\Psi(0)\rangle, \tag{105}$$

$$\tilde{F}_{l,l'} = \langle\Psi(0)|\begin{pmatrix} c_{2l-1} \\ c_{2l} \end{pmatrix}(c_{2l'-1}, c_{2l'})|\Psi(0)\rangle. \tag{106}$$

We have

$$\tilde{C}_{l,l'} = \frac{(-1)^{l'-l}}{4}\sin^2\theta\cos(\theta)^{2(l'-l)-1}\begin{pmatrix} 1 & -\cos(\theta) \\ 1/\cos(\theta) & -1 \end{pmatrix}, \tag{107}$$

and $\tilde{C}_{l',l} = \tilde{C}_{l,l'}^\dagger$ if $l' > l$, while

$$\tilde{C}_{l,l} = \begin{pmatrix} \sin^2(\theta/2) & -\frac{1}{4}\sin^2\theta \\ -\frac{1}{4}\sin^2\theta & \cos^2(\theta/2) \end{pmatrix}. \tag{108}$$

On the other hand, the matrices $\tilde{F}_{l,l'}$ read

$$\tilde{F}_{l,l'} = \frac{(-1)^{l'-l}}{4}\sin^2\theta\cos(\theta)^{2(l'-l)-1}\begin{pmatrix} -1 & \cos\theta \\ -1/\cos\theta & 1 \end{pmatrix}, \tag{109}$$

and $\tilde{F}_{l',l} = -\tilde{F}_{l,l'}^t$ for $l' > l$, and

$$\tilde{F}_{l,l} = \begin{pmatrix} 0 & \frac{1}{4}\sin^2\theta \\ -\frac{1}{4}\sin^2\theta & 0 \end{pmatrix}. \tag{110}$$

Observe that, since the matrices $\tilde{C}_{l,l'}$ and $\tilde{F}_{l,l'}$ only depend on the difference $l'-l$, they can be seen as the Fourier coefficients of certain $2 \times 2$ matrices $\tilde{C}$ and $\tilde{F}$ defined on the unit circle; that is,

$$\tilde{C}_{l,l'} = \int_{-\pi}^{\pi} \frac{dk}{2\pi} \tilde{C}(k) e^{-ik(l-l')}, \quad \tilde{F}_{l,l'} = \int_{-\pi}^{\pi} \frac{dk}{2\pi} \tilde{F}(k) e^{-ik(l-l')}. \tag{111}$$

The matrices $\tilde{C}(k)$ and $\tilde{F}(k)$ can be determined by performing the inverse Fourier transform in Eq. (111)

$$\tilde{C}(k) = \sum_{l-l'\in\mathbb{Z}} \tilde{C}_{l,l'} e^{ik(l-l')}, \quad \tilde{F}(k) = \sum_{l-l'\in\mathbb{Z}} \tilde{F}_{l,l'} e^{ik(l-l')}. \tag{112}$$

The series above can be explicitly calculated with e.g. Mathematica and one eventually obtains

$$\tilde{C}(k) = \frac{1}{2} \begin{pmatrix} 1 + g_{11}(k,\theta) & e^{ik/2} g_{12}(k,\theta) \\ e^{-ik/2} g_{12}(k,\theta) & 1 - g_{11}(k,\theta) \end{pmatrix}, \tag{113}$$

which gives the diagonal blocks of the symbol of Eq. (23), and

$$\tilde{F}(k) = \frac{i}{2} \begin{pmatrix} f_{11}(k,\theta) & e^{ik/2} f_{12}(k,\theta) \\ e^{-ik/2} f_{12}(k,\theta) & -f_{11}(k,\theta) \end{pmatrix}, \tag{114}$$

which corresponds to the off-diagonal blocks of Eq. (23).

We pass now to calculate the correlation matrices $\langle \Psi(t)| c_j^\dagger c_{j'} |\Psi(t)\rangle$ and $\langle \Psi(t)| c_j c_{j'} |\Psi(t)\rangle$ after the quench to the XX spin chain Hamiltonian (9). In this case, it is useful to change to the Heisenberg picture. In fact, since the post-quench Hamiltonian is diagonalised by a Fourier transformation, the fermionic operators in momentum space $c_k$ trivially evolve in time as $c_k(t) = e^{-it\epsilon_k} c_k$. Therefore, the time dependence of the fermionic operators in the real space $c_j$ is given by

$$c_j(t) = \sum_{j'\in\mathbb{Z}} R_{jj'}(t) c_{j'}, \quad R_{jj'}(t) = \int_{-\pi}^{\pi} \frac{dk}{2\pi} e^{-it\epsilon_k} e^{-ik(j-j')}. \tag{115}$$

If we consider the $2 \times 2$ matrices $\tilde{C}_{l,l'}(t)$ and $\tilde{F}_{l,l'}(t)$ analogous to the ones in Eq. (111) but for the state $|\Psi(t)\rangle$ and we apply Eq. (115), then we find

$$\tilde{C}_{l,l'}(t) = \int_{-\pi}^{\pi} \frac{dk}{2\pi} \mathcal{R}(k,t)^\dagger \tilde{C}(k) \mathcal{R}(k,t) e^{-ik(l-l')}, \tag{116}$$

and

$$\tilde{F}_{l,l'}(t) = \int_{-\pi}^{\pi} \frac{dk}{2\pi} \mathcal{R}(k,t) \tilde{F}(k) \mathcal{R}(k,t) e^{-ik(l-l')}, \tag{117}$$

with

$$\mathcal{R}(k,t) = \begin{pmatrix} \cos(t\epsilon_{k/2}) & i e^{ik/2} \sin(t\epsilon_{k/2}) \\ i e^{-ik/2} \sin(t\epsilon_{k/2}) & \cos(t\epsilon_{k/2}) \end{pmatrix}. \tag{118}$$

Observe that, when we apply Eq. (115) to calculate $\tilde{C}_{l,l'}(t)$ and $\tilde{F}_{l,l'}(t)$, the entries $R_{j,j'}(t)$ must be rearranged according to the two-site translational invariance of the initial state $|\Psi(0)\rangle$. This has the effect that the dispersion relation of $H$ enters in $\mathcal{R}(k,t)$ as $\epsilon_{k/2}$ instead of $\epsilon_k$. One can check that the matrices $\mathcal{C}_t(k,\theta)$ and $\mathcal{F}_t(k,\theta)$, introduced respectively in Eqs. (29) and (31), are precisely $\mathcal{C}_t(k,\theta) = 2\mathcal{R}(k,t)^\dagger \tilde{C}(k) \mathcal{R}(k,t) - I$ and $\mathcal{F}_t(k,\theta) = 2\mathcal{R}(k,t) \tilde{F}(k,\theta) \mathcal{R}(k,t)$.

# D Derivation of Eqs. (85) and (86)

We can use the results of Appendix C to get the correlators of Eqs. (85) and (86). In particular, to obtain the expression (85) we need to compute

$$\langle \Psi(0)| c_k^\dagger c_k |\Psi(0)\rangle = \sum_{j-j'\in\mathbb{Z}} e^{ik(j-j')} \langle \Psi(0)| c_j^\dagger c_{j'} |\Psi(0)\rangle . \tag{119}$$

Taking into account Eqs. (107) and (108), we can split the previous sum in even and odd sites,

$$\langle \Psi(0)| c_k^\dagger c_k |\Psi(0)\rangle = \frac{1}{2} + \frac{1}{2} \sum_{l-l'\in\mathbb{Z}} e^{2ik(l-l')} \Big[ e^{-ik} \langle \Psi(0)| c_{2l-1}^\dagger c_{2l'} |\Psi(0)\rangle$$
$$+ e^{ik} \langle \Psi(0)| c_{2l}^\dagger c_{2l'-1} |\Psi(0)\rangle \Big], \tag{120}$$

where we applied that $\langle \Psi(0)| c_{2l-1}^\dagger c_{2l'-1} |\Psi(0)\rangle = -\langle \Psi(0)| c_{2l}^\dagger c_{2l'} |\Psi(0)\rangle$ for $l \neq l'$. Replacing the remaining correlators by their explicit expressions in Eq. (107), we have

$$\langle \Psi(0)| c_k^\dagger c_k |\Psi(0)\rangle = \frac{1}{2} - \frac{\sin^2(\theta)\cos(k)}{4}$$
$$+ \frac{1}{8} \sum_{j=-\infty}^{-1} e^{2ikj}(-1)^j \sin^2(\theta)\cos^{-2j-1}(\theta)\big[ e^{ik}\sec(\theta) - e^{-ik}\cos(\theta) \big]$$
$$+ \frac{1}{8} \sum_{j=1}^{\infty} e^{2ikj}(-1)^j \sin^2(\theta)\cos^{2j-1}(\theta)\big[ e^{-ik}\sec(\theta) - e^{ik}\cos(\theta) \big]. \tag{121}$$

The series above yield Eq. (85),

$$\langle \Psi(0)| c_k^\dagger c_k |\Psi(0)\rangle = \frac{1}{2}\left( 1 - \frac{\cos(k)(1-\cos^4\theta)}{1+2\cos(2k)\cos^2(\theta)+\cos^4(\theta)} \right). \tag{122}$$

By applying the same steps as before, we can obtain Eq. (86). Let us take

$$\langle \Psi(0)| c_{\pi-k}c_k |\Psi(0)\rangle = \sum_{j-j'\in\mathbb{Z}} e^{ik(j-j')-i\pi j} \langle \Psi(0)| c_j c_{j'} |\Psi(0)\rangle , \tag{123}$$

and distinguish even and odd sites, then

$$\langle \Psi(0)| c_{\pi-k}c_k |\Psi(0)\rangle = \frac{1}{2} \sum_{l-l'\in\mathbb{Z}} e^{2ik(l-l')}\big[ -\langle \Psi(0)| c_{2l-1}c_{2l'-1} |\Psi(0)\rangle$$
$$+ \langle \Psi(0)| c_{2l}c_{2l'} |\Psi(0)\rangle \big], \tag{124}$$

where we took into account that $\langle \Psi(0)| c_j c_{j'} |\Psi(0)\rangle = -\langle \Psi(0)| c_{j'} c_j |\Psi(0)\rangle$. If we substitute in this equation the correlators by their explicit expressions given in Eqs. (109) and (110),

$$\langle \Psi(0)| c_{\pi-k}c_k |\Psi(0)\rangle = \frac{1}{4} \sum_{j=-\infty}^{-1} e^{2ikj}(-1)^j \sin^2(\theta)\cos^{-2j-1}(\theta)$$
$$- \frac{1}{4} \sum_{j=1}^{\infty} e^{2ikj}(-1)^j \sin^2(\theta)\cos^{2j-1}(\theta), \tag{125}$$

we finally find Eq. (86)

$$\langle \Psi(0)| c_{\pi-k}c_k |\Psi(0)\rangle = \frac{i}{2}\frac{\sin(2k)\sin^2(\theta)\cos(\theta)}{1+2\cos(2k)\cos^2(\theta)+\cos^4(\theta)}. \tag{126}$$

# E  Determinant involving a product of block Toeplitz matrices

One of the main results on the theory of block Toeplitz determinants is the Widom-Szegő theorem [34]. According to it, the determinant of a block Toeplitz matrix $T_\ell[g]$ with symbol $g$, see Eq. (35), behaves for large $\ell$ as

$$\log \det T_\ell[g] \sim \ell \int_0^{2\pi} \frac{dk}{2\pi} \log \det g(k), \tag{127}$$

if $\det g[k] \neq 0$ with zero winding number.

If $T_\ell[g]$ and $T_\ell[g']$ are two arbitrary block Toeplitz matrices with a symbol given by, respectively, $g(k)$ and $g'(k)$, then the entries of the product $T_\ell[g]T_\ell[g']$ behave in the limit $\ell \to \infty$ as

$$
\begin{aligned}
(T_\ell[g]T_\ell[g'])_{ll'} &= \sum_{j=1}^{\ell} (T_\ell[g])_{lj}(T_\ell[g'])_{jl'} \\
&\sim \sum_{j=1}^{\infty} (T_\ell[g])_{lj}(T_\ell[g'])_{jl'} = \int_0^{2\pi} \frac{dk}{2\pi} g(k)g'(k) e^{-ik(l-l')},
\end{aligned} \tag{128}
$$

where we have extended the sum over $j$ until $\infty$ and we have used $\sum_{j=1}^{\infty} e^{i(k-k')j} = 2\pi\delta(k-k')$. Thus, from Eq. (128), we may conclude that the entries of the product $T_\ell[g]T_\ell[g']$ behave as the ones of the block Toeplitz matrix $T_\ell[gg']$ in the limit $\ell \to \infty$. This observation, combined with Widom-Szegő theorem, allows to derive the asymptotic behaviour with $\ell$ of the determinant of matrices of the form $I + T_\ell[g]T_\ell[g']$, which are relevant in the analysis performed in this paper. In fact, according to it, $\det(I + T_\ell[g]T_\ell[g']) \sim \det(I + T_\ell[gg'])$ and applying Widom-Szegő theorem of Eq. (127) we conclude that

$$\log \det(I + T_\ell[g]T_\ell[g']) \sim \ell \int_0^{2\pi} \frac{dk}{2\pi} \log \det(1 + g(k)g'(k)). \tag{129}$$

This result also extends to the product of more than two block Toeplitz matrices, as we do in Eq. (36).

The conjecture of Eq. (38) involving the inverse of the Toeplitz matrices $T_\ell[g'_j]$ can be also derived as a corollary of Eq. (37). Indeed, if we take $n = 1$ and we reexpress $T_\ell[g']$ as $T_\ell[g'] = I - T_\ell[I - g']$, then, by using the power series

$$(I - T_\ell[I - g'])^{-1} = \sum_{p=0}^{\infty} T_\ell[I - g']^p, \tag{130}$$

the left hand side of Eq. (38) can be recast into Eq. (37) as

$$\det\left(I + T_\ell[g]T_\ell[g']^{-1}\right) = \det\left(I + \sum_{p=0}^{\infty} T_\ell[g]T_\ell[I - g']^p\right). \tag{131}$$

Given that the sum of Toeplitz matrices is still a Toeplitz matrix, we can apply Eq. (128) to deduce that, for large $\ell$,

$$\det\left(I + T_\ell[g]T_\ell[g']^{-1}\right) \sim \det\left(I + T_\ell\left[\sum_{p=0}^{\infty} g(I - g')^p\right]\right). \tag{132}$$



Taking into account that $(g')^{-1} = \sum_{p=0}^{\infty} (I - g')^p$, we have

$$\det\left(I + T_\ell[g]T_\ell[g']^{-1}\right) \sim \det\left(I + T_\ell\left[g(g')^{-1}\right]\right), \tag{133}$$

and, by virtue of the Widom-Szegő theorem (127),

$$\log\det\left(I + T_\ell[g]T_\ell[g']^{-1}\right) \sim \ell \int_0^{2\pi} \frac{dk}{2\pi} \log\det\left(I + g(k)g'(k)^{-1}\right), \tag{134}$$

in agreement with our conjecture (39).

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
