# Peer review of "Lack of symmetry restoration after a quantum quench: an entanglement asymmetry study"

_SciPost Physics, doi:SciPost Phys. 15, 089 (2023)_

## Round 2 · Referee Report · Anonymous (Referee 1) · 2023-4-20

Strengths

1- A new concept of symmetry breaking of quantum mechanics states
2-A case study that complements the intial ones.
3- A surprise results, a its possible explanation
4- Very acurate and clear paper.

Report

Title: Lack of symmetry restoration after a quantum quench: an
entanglement asymmetry study

Authors: F. Ares, S. Murciano, E. Vernier and P. Calabrese

Some of the authors of the present paper introduced recently an interesting
measure of symmetry braking of quantum states defined in the Hilbert space
spanned in quantum spin chains [5]. They proposed a measure of this symmetry in
terms of the reduced entanglement entropy continuous subsystems, after a projection of the non-symmetric state into a symmetric one. They call this measure as
entanglement asymmetry. The results in was derived from initial states which
are cat states created from an uniform ferromagnetic one, and consider its
time evolution governed by the standard XX quantum chain. The results shoes
that at larger times the U(1) symmetry not present in the initial state is
restored. Interestingly the time to reach the symmetry decreases with the
increase of the asymmetry of the initial state, resembling a quantum version of the Mpemba classical effect.
In the present paper they study a different class of cat states formed from a
Neel anti ferromagnetic state, again with the dynamics ruled by the XX quantum.
Surprisingly the evolution dos not restore in general the U(1) symmetry of the
time evolution operator, reaching an stationary state non symmetric.
Only for the initial state with the largest asymmetry the symmetry is restored
at asymptotic times. They also analyze these stationary states and give a
possible explanation in terms of the presence of non-abelian conserved charges
presented in the initial state.

Their results, analytical and numerical are carefully presented. The
mathematics involved is the extension of known ones, with also some
interesting conjectures. The paper is very well writtem and with a clear
presentation. I suggest its acceptance in the journal.

---

## Round 2 · Referee Report · Anonymous (Referee 2) · 2023-5-4

Strengths

1- Clear and concise review of the definitions and properties associated
with entanglement asymmetry, and of the relevant results from the
authors' previous work.
2- The main observations and statements of the manuscript are appealing
and novel, and are also easy to understand.
3- The authors identify a general mechanism behind the results that they
have derived, allowing to put them in a wider context related to a
timely subject of research.

Weaknesses

1- Many important steps of the derivation are obscured from the reader,
making it hard to apply the method to other similar problems and to
independently verify the validity of the results.
2- The manuscript seems to contain a non-negligible number of discrepancies,
which originate either in typos or in the absence of a clear explanation
on how a certain formula was exactly used to derive another formula.

Report

In this manuscript, the authors present an analytical study of the
dynamics of the XX spin chain (that can be mapped to 1D free fermions),
focusing on the competition between the Hamiltonian symmetry that
conserves magnetization (or fermionic charge) and the breaking of
that symmetry by the initial state in which the system is prepared.
The authors use a measure of symmetry-breaking (in a subsystem of
an extended system) defined by them in a previous work (Ref. [5]),
called "entanglement asymmetry". They apply a combination of rigorous
arguments and sound conjectures to derive asymptotic formulae for
the time evolution of this measure, which they then verify numerically.
Their main finding is that when the system is initially prepared in
a tilted anti-ferromagnetic state, symmetry is not restored even in
the long-time limit, in stark contrast with the case of a tilted ferromagnetic
initial state studied in Ref. [5]. The authors present an elegant
and compelling explanation of the origin of this difference, relating
it to a non-Abelian set of conserved charges that is trivial (with
respect to the symmetry in question) in the latter case but not in
the former.

In my opinion, the manuscript deals with a timely topic and showcases
the strengths of entanglement asymmetry in unveiling new insights
on nonequilibirium quantum dynamics. It also nicely sets the stage
for further studies on interacting models. However, I believe that
major issues with the presentation must be addressed before I can
fully recommend the manuscript for publication in SciPost Physics.
The authors seem to have done serious analytical work, but many details
of the derivation of their results, including those of some crucial
steps, are obscured from the reader. As I am mostly referring to technical
aspects of the different calculations, I do not expect my comments
to lead to an extreme overhaul of the main text itself, but rather
to the addition of some appendices that would clarify these points.

I detail my specific comments in the "list of changes" attached
below. Yet to emphasize the need for the revision I am requesting,
I highlight here two points in the manuscript where, in my opinion,
this lack of details is particularly salient. I am referring to the
claim that the initial state defined in Eq. (6) is Gaussian, and to
the explicit expressions for the correlation matrix in Eqs. (22),
(28) and (30). No proof is shown for the first claim (this might be
a known result of which I am not aware, but then a proper reference
should be cited), and no calculation is shown for these explicit expressions,
even though they form the basis for the entire analysis. Moreover,
as they form the shared starting point for both analytical *and*
numerical calculations, they cannot be a posteriori justified
through a comparison between analytical and numerical results; the
process of their derivation must be clearly presented to establish
the validity of the ensuing analysis. Just as a simple example of
a problem resulting form this issue, I will note that from the expression
in Eq. (22) for the initial-state correlation matrix it appears that,
as the tilt-angle $\theta$ tends to $0$, the average occupation
of odd-index fermionic sites tends to $0$, even though from the definition
in Eq. (7) it appears that it should tend to $1$; this might result
from a typo, but as no part of the calculation process is presented,
it was impossible for me to trace the source of this apparent discrepancy.

Additional specific comments appear in the list of requested changes.
While I was highly intrigued by the results reported in this manuscript,
and enjoyed the clarity of the qualitative observations the authors
had brought forward, I cannot wholeheartedly attest to their full
validity before I can examine more details of the derivation. I hope
my comments below serve to improve the presentation and technical
clarity of the manuscript, as well as to resolve some more minor issues
I found while reading it.

Requested changes

Missing details:
1- Page 4: There is no explanation (or cited reference) substantiating
the statement that the chosen initial state in Eq. (6) is Gaussian.
This is a crucial ingredient in the remainder of the analysis, and
thus it should be explained.
2- Page 4: The authors state that the entanglement asymmetry is identical
for the tilted ferromagnetic and tilted N\'eel states, and explain
this only through the phrase "because of the product-state structure".
This explanation is rather too concise, given that this statement
is not merely ornamental, as it is crucially utilized later in deriving
the ansatz in Eq. (45). This explanation is also a bit confusing as
the cat versions of both states are not product states. I believe
a very brief mathematical argument/proof can be added here to show
this equality.
3- Page 6: The derivation of Eq. (14) is not explicitly shown. I am aware
that this formula already appeared in Ref. [5] by the authors,
but also there they only refer to previous works that established
general composition rules of Gaussian operators, without showing how
they used them directly in this specific case. Since the formula in
Eq. (14) is both an important novelty of the authors' work (it can
be used in other similar studies) and a crucial basis for all the
(analytical *and* numerical) calculations in this manuscript,
I believe it would be useful if the authors briefly demonstrated how
this formula came about (namely: what is the form of the composition
rule that was used here).
4- Pages 7-8: The derivation of Eqs. (22), (28) and (30) is not trivial
and should be detailed. In particular, writing the correlations at
$t=0$ in the form of a $k$-space integral is not a straightforward
step, as the initial state is not conspicuously related to the energy
eigenstates. As I emphasized in the main part of the report, the correlation
matrix is a crucial component in the calculations of all the results
appearing in the manuscript, and thus the process of its computation
should be made more accessible to the reader. This pertains in particular
to the my suspicion of a typo that I mentioned, which might amount
only to an erroneous minus sign on some (or all) of the matrix terms,
though this cannot be checked without more details. Another important
feature of these expressions, which is not addressed in the text,
is the appearance of $\epsilon_{k/2}$ (rather than $\epsilon_{k}$)
in the time evolution of the matrix elements; this detail also comes
back later when the appropriate quasi-particle velocity is determined.
I suppose that this arises due to the reduced periodicity of the initial
state, but again it is not trivial to see, and the derivation should
be written down.
(I will note here that, compared to the definition of $n_{+}\!\left(k,\theta\right)$
in page 10, the authors seem to have flipped the sign of $f_{11}\!\left(k,\theta\right)$
and $g_{12}\!\left(k,\theta\right)$ and substituted $2k$ rather than $k$ when $n_{+}\!\left(k,\theta\right)$ reappears
in Eq. (86); this adds to the confusion regarding the explicit expressions
for the correlation matrix elements).
5- Page 10: Starting from Eq. (39), a new parameter $\alpha$ appears,
and it should be clearly stated that it is defined as $\alpha=\alpha_{1}-\alpha_{2}$.
6- Page 11: It is not immediately clear why excitations with momentum
$k$ propagate at a group velocity of $\left|\epsilon_{k/2}'\right|$
rather than $\left|\epsilon_{k}'\right|$; the authors should elaborate
on this point (this also goes back to item no. 4 in this list).
7- Page 14: The authors show how to derive Eq. (64) for $n=2$, and write
that for $n>2$ the general formula can be derived through "a similar
reasoning". However, the case $n=2$ is rather special, in that
the integral that transforms the charged moment into the entanglement
asymmetry reduces to a single-variable integral, while for $n>2$
a multi-variable saddle-point approximation (which is less straightforward
than the single-variable one) must be invoked. I therefore believe
that it would be beneficial if an explicit calculation for general
$n$ was shown here (such a general calculation is possible, given
that the authors report its final result). The need for such an explicit
demonstration of the calculation is underlined by the fact that an
equivalent general formula *cannot* be obtained for $t\to\infty$.
The authors state that they could not obtain a closed expression for
$g_{\infty}^{\left(n\right)}$ in Eq. (68), but that for each $n$
this term can be reconstructed using the multi-variable saddle-point
analysis; in my opinion, this calls for a demonstration of this analysis
(based on which $g_{\infty}^{\left(n\right)}$ can be straightforwardly
reconstructed), at least in the case of $t=0$ where the general formula
is within reach.
8- Page 16: The plots in Fig. 5 test the agreement of the analytical
result for $t\gg\ell$ with numerics, but for the chosen fixed value
of $t$ the agreement is clear only for $\ell$ of the order of $t$.
This is a bit confusing, but from the right panel of Fig. 3 it seems
that the chosen value $t=200$ is above the saturation time of the
asymmetry, at least for the values of $\theta$ for which results
are presented in Fig. 5. The authors should mention that (or provide
another explanation), otherwise it is not clear why the numerical
results in Fig. 5 for $\ell\sim170$ and $t=200$ indeed capture the
steady-state limit (the limit at which the analytical results were
computed).
9- Page 18-19: Deriving the initial expectation values in Eqs. (79)
and (80) is not a trivial task, again due to the fact that the initial
state is not simply defined through a certain occupation function
of energy eigenstates. The origin of these expressions might be clarified
once the authors add details on the derivation of the correlation
matrix in Eq. (22). In any case, it would be good if the authors could
briefly elaborate on the derivation of these expressions, given that
they are at the heart of the argument they present in Sec. 5.

Apparent discrepancies or typos:
1- Page 3: "iif" should be "iff".
2- Page 14: As far as I understood from the authors' claim on page 4,
the entanglement asymmetry of the initial tilted N\'eel state in
Eq. (64) should be equal to that of the tilted ferromagnetic state,
given in Eq. (9) of Ref. [5]. However, this is not the case unless
$g_{0}\!\left(\theta\right)=2\sin^{2}\!\left(\theta\right)$ for all
$\theta\neq0,\pi$, and it can be directly checked through Eqs. (47)
and (61) that, e.g., $g_{0}\!\left(\theta=\pi/2\right)=1/2$. This
should be resolved (or clarified, if for some reason one should not
expect Eq. (64) to exactly reproduce Eq. (9) of Ref. [5]).
3- Page 15: In the caption of Fig. 4, "the prediction (68)" should
be written instead of "the prediction (67)".
4- Page 17: A factor of $1/2$ should multiply the sums over $j$ in
Eqs. (70) and (71). A minus sign should appear before the integral
in Eq. (70).
5- Page 18: By examining the first-order term in the expansion of ${\cal U}_{k}c_{k}{\cal U}_{k}^{\dagger}$
with ${\cal U}_{k}$ defined in Eq. (77), it appears that a factor
of $i$ is missing from the off-diagonal terms of the matrix in Eq.
(76), which is supposed to capture the same Bogoliubov rotation.
6- Page 18: A minus sign should appear in the LHS of Eq. (78), by comparison
to Ref. [5].
7- Page 19: Strictly speaking, if $r_{k},\varphi_{k}$ are chosen to
be constant in $k$ as is done before Eq. (81), then the expression
obtained for $c_{\pi-k}$ from Eq. (76) by substituting $\pi-k$ instead
of $k$ in the first row is inconsistent with the conjugate of $c_{\pi-k}^{\dagger}$
from the second row of Eq. (76). This can be resolved if in Eq. (76)
$k$ is limited to the domain $\left[-\frac{\pi}{2},\frac{\pi}{2}\right]$.
8- Page 19: The $i$ in Eq. (83) should appear with an opposite sign.
9- Page 20: A minus sign is missing in the exponent appearing in Eq.
(88).

Further questions and suggestions:
1- Page 9: I believe that the determinant asymptotics of Eq. (37) can
be formulated as a corollary of the conjecture Eq. (35) rather than
as a separate conjecture. The inverted Toeplitz matrices that appear
in the analysis are of the form $\left(I-\Gamma\right)^{-1}$, and
can therefore be formally written as a power series in $\Gamma$;
in such a way, the product in the LHS of Eq. (37) becomes an (infinite)
sum of products of the form that appears in the LHS of Eq. (35); employing
Eq. (35) (considering that a sum of Toeplitz matrices *is* a
Toeplitz matrix) and summing again the series, I expect that Eq. (37)
is obtained, given that $I-g$ is the symbol of $I-\Gamma$ if $g$
is the symbol of $\Gamma$. This might seem more appealing than stating
that Eq. (37) is a new conjecture altogether, but I leave this to
the consideration of the authors.
2- Page 10: Can the authors comment on why they necessarily chose the
factorization in Eq. (46)? An agreement with the $n=2$ case in Eq.
(41) is obtained also, e.g., if $f_{k}^{{\rm N}}=\lambda\sin\!\left(\alpha\right)+\cos\!\left(\alpha\right)$
for an appropriate choice of $\lambda$ (this looks more like Eq.
(18)).
3- Page 13: In the right panel of Fig. 2 there is a peculiar feature
that the authors do not address. For $\theta=4/5$ and $t/\ell\sim0.5$,
it seems like the numerical results move *away* from the analytical
prediction as $\ell$ increases, while they are expected to converge
to it. Can the authors explain the source of this result?
4- Page 18: Right after Eq. (78), the authors write that "plugging
the above expression in the quasi-particle picture for the entanglement
asymmetry yields the correct result". It is not entirely clear,
however, to what specific formula in terms of $n\!\left(k\right)$
they are referring as the formula into which the expression for $n\!\left(k\right)$
in Eq. (78) should be substituted (maybe it is the regular formula
for the entanglement entropy, rather than the entanglement asymmetry?).
It would be helpful if they can point to the appropriate equation
(in the manuscript or in Ref. [5]), or write down this equation
if it is not written elsewhere. I think it will better illustrate
the authors' argument regarding the generality of their GGE treatment.

  • validity: -
  • significance: high
  • originality: high
  • clarity: -
  • formatting: -
  • grammar: reasonable

Author:  Sara Murciano  on 2023-06-06  [id 3709]

(in reply to Report 2 on 2023-05-04)

We greatly thank the referee for their careful reading of the paper. We believe that, thanks to their several comments and suggestions, we have improved the quality and clarity of the manuscript by adding more details about the derivation of the results, especially in the four new appendices. We would like to start by pointing out that the discrepancy between the initial-state correlation matrix, Eq. (23) in the new version of the manuscript, and Eq. (7) is due to a typo in the ordering in which the up and down spins were written in Eq. (7), which we have corrected. In what follows, we provide a point-by-point reply. Given the several modifications we have done to our draft, all the equations in our reply refer to the numbering of the present version of the manuscript.

Missing details:

1- We thank the referee for this question. We have added Appendix A where we explicitly prove that the reduced density matrix of the cat version in Eq. (6) of the tilted N\'eel state is Gaussian by showing that it is an eigenstate of a Hamiltonian which is quadratic in the fermionic operators $ c^{\dagger}_j$, $c_j$.

2- We thank the referee for their comment. We have removed that sentence because it is actually not correct: the Rényi entanglement asymmetries of the tilted Néel and ferromagnetic states, although very close, are not exactly equal except in the large subystem limit $\ell\to\infty$. We have added the explicit result when $\ell\to \infty$ for the asymmetry of the cat tilted ferromagnetic state in the new Eq. (20) and we have remarked that it slightly differs from the one of the tilted ferromagnetic state reported in Ref. [5], as the referee also points out. Furthermore, after deriving the analogous result for the cat tilted Néel state in Eq. (69), we now compute explicitly in Eq. (70) the integral that gives the term $g_0(\theta)$ and show that the asymptotic behaviour of the Rényi entanglement asymmetry is the same for the cat tilted ferromagnetic and N\'eel states in the large interval limit. When deriving the ansatz of Eq. (46), we have removed the argument on the equality of the entanglement asymmetries of the tilted Néel and ferromagnetic states. We now write that, inspired by the factorization in the replica index found for the tilted ferromagnet, one can conjecture that the charged moments of the cat tilted Néel state admit a similar decomposition, an ansatz that we have checked numerically as shown in the paper. 3- Given the relevance of Eq. (14), both for the analytical and numerical computations of the manuscript, we have added its analytical derivation in Appendix B. 4- In the new version of the manuscript, we have added Appendix C where we explicitly derive the two-point correlators reported in subsection 3.1 for the cat tilted N\'eel state in Eq. (6) and its time evolution (8) after the quench to the XX spin chain. We think that, from the calculation presented in the new appendix, it is clear that the appearance of $\epsilon_{k/2}$ rather than $\epsilon_k$ in the time-evolved correlators is originated by the arrangement of the correlation matrix $\Gamma(t)$ as a block Toeplitz matrix due to the two-site periodicity of the initial state. We have also emphasized this point by adding a comment in the main text after Eq. (33) and before Eq. (53). We have also fixed the flipped signs in Eq. (92) and indicated after that equation that $\widetilde{n}$ identifies with $n_-$ upon replacing $k\mapsto 2k$ in the latter. The reason is that $\widetilde{n}$ has been calculated using the modes $c_k$, $c_k^\dagger$ that diagonalize the post-quench Hamiltonian $H$ while in the calculation of $n_{-}$ the Brillouin zone of $H$ is halved. 5- We have added the definition of $\alpha$ after Eq. (40). 6- As mentioned in the point 4, we think that both the derivation in Appendix C and the comment before Eq. (53) clarify this point. 7- We report after Eq. (64) the explicit computation of the Rényi entanglement asymmetry at $t=0$ for any integer $n$ using the multi-variable saddle-point. The approach is similar for $t\to \infty$, the only difference is that the charged moments are of the form given by Eq. (50), which does not factorize in the replica index as at $t=0$, Eq. (46). This implies that the term $g_\infty^{(n)}(\theta)$ has a different expression for each integer $n$ and it has to be worked out case by case. 8- We have added a comment at the end of Section 4 regarding Figs. 3 and 5: from the right panel of Fig. 3, we deduce that $t=200$ actually corresponds to the saturation regime of the asymmetry. However, since our results hold in the limit $\ell\to \infty$, a good agreement is visible only for large subsystem size $\ell$. 9- We have included Appendix D where we present the derivation of the mode occupation numbers of Eqs. (85) and (86) by evaluating the various bilinear combinations of fermionic operators in the initial state.

Apparent discrepancies or typos:

1- We have corrected the typo. 2- As mentioned in a previous point, the entanglement asymmetries of the tilted Néel and ferromagnetic states and their cat versions are slightly different. Eq. (9) of Ref. [5] corresponds to the asymptotic behaviour of the asymmetry for the (non-cat) tilted ferromagnet. We have added in Eq. (20) the analogous result for its cat version and showed after Eq. (69) that the cat tilted Néel state presents the same asymptotic behaviour. We have emphasized that the asymmetry of the tilted Néel and ferromagnetic states and their cat versions yield a slightly different result after Eq. (20) and Eq. (70). 3- We have corrected this typo. 4- We have fixed the various sign errors in agreement with the referee's suggestion (note in particular that the sign of $H_m$ has been changed with respect to our previous definition). 5- We thank the referee for this observation. This was due to an extra factor of $i$ in the definition of the operator $\mathcal{U}_k$. We have corrected the latter, and kept the action (81) of the Bogoliubov rotation intact. 6- We have corrected the sign typo. 7- We agree with the referee's comment and have changed the range of $k$ to $[-\pi/2,\pi/2]$. 8- We have corrected the sign typo. 9- We have corrected the sign typo in the current Eq. (128).

Further questions and suggestions

1- We thank the referee for the useful comment, which we think it is correct. We have included in Appendix E a proof that Eq. (38) can be formulated as a corollary of the conjecture of Eq. (36). 2- We thank the referee for the suggestion. However, we were not able to reproduce with the suggested expression for $f_{k}^{{\rm N}}$ the correct result for the charged moments of the entanglement asymmetry when $n>2$. Indeed, it would produce a complex result already at $n=3$, while we have checked numerically that it is real. We remark that the structure of the factorization is very similar to the one that we find for the tilted ferromagnetic state in Eqs. (16), (17), but not in the explicit form of the function $f_k^{\rm N}$. 3- We have added a comment at the end of Section 3. For $n=3$, $\theta=4/5$ and $\ell=100$, the numerical point in Fig. 2 around $t/\ell=0.5$ deviates from the analytical prediction due to numerical errors originated in the calculation of the inverse matrix $(I-\Gamma(t))^{-1}$ that appears in Eq. (14). These errors become relevant for large $\ell$ in the regions where $Z_n(\boldsymbol{\alpha}, t)$ presents a peak like in this case. 4- We thank the referee for pointing out this lack of clarity in our original text. We have now provided a new explanation, linking to the explicit formulae using the quasi-particle picture.

---

## Round 3 · Referee Report · Anonymous (Referee 2) · 2023-6-19

Report

In my opinion, the revised version of the manuscript presents a considerable
improvement over its original version. My previous comments regarding
the conceptual strengths of the manuscript obviously still hold, and
now the technical aspects of the work have been made much clearer
owing to the details and appendices added by the authors. The authors
have largely answered my questions, requests and concerns and I thank
them for their efforts.

In my reading of the new version I still found some minor technical
issues and typos that should be addressed, which I included in the
subsequent list of requested changes. In addition, I begin the list
with a question to the authors on the physics side of things, namely
regarding the identification of an "Mpemba effect" similar to
the one they have observed in their previous work.

I would recommend the manuscript for publication once the remaining
issues are resolved.

Requested changes

1- When reviewing their previous results on the quench from a tilted ferromagnetic initial state, the authors mention a striking property to which they refer as a "quantum Mpemba effect", where the rate at which the initially-broken symmetry is restored increases as the entanglement asymmetry of the initial state increases. A similar effect seems to be captured by the right panel of Fig. 3; though the symmetry is not eventually restored and the asymmetry does not decay to 0, the rate at which the asymmetry drops towards its saturation value increases with increasing asymmetry at $t=0$. Could the authors comment whether this observation is consistent with all the numerical checks they performed? If this is indeed an effect that applies also for the N\'eel quench, I believe it would be useful for the reader if this was emphasized in the text. If so, are the authors able to provide some analytical insight regarding the effect, for example through the quasiparticle picture that they present (I am thinking, for example, on a possible argument where one shows that, as we vary $\theta$, the contribution of a quasiparticle with momentum $k$ to the integral defining $B_{n}\left(\alpha,\zeta\right)$ peaks at a different $k=k^{*}\left(\theta\right)$, and that the group velocity associated with this $k^{*}$ increases with increasing $\theta$)?

2- On page 5, the authors write that, for the initial state they examine, $\Delta S_{A}$ is monotonic in $\theta$, while it is not so for all $n$ when considering $\Delta S_{A}^{\left(n\right)}$. Could they mention the source of this claim (numerical checks on a finite interval, etc.)? Their result for large $\ell$ in Eq. (69) suggests that $\Delta S_{A}^{\left(n\right)}$ is always monotonic, so I assume this claim refers to the finite $\ell$ case, but this should be clarified in any case.

3- The definition of $n_{A}$ appears twice: once on page 7 after Eq. (14), and once on page 23 after Eq. (96). The signs are flipped between these two definitions, so this should be fixed.

4- On page 10, in Eq. (34), one entry of the matrix says $g_{12}\left(k\right)$ instead of $g_{12}\left(k,\theta\right)$. This typo should be fixed.

5- On page 16, there are typos in Eq. (67): inside the exponent, $l$ should be replaced with $\ell$, and $\beta_{j}$ (in the first sum) should be replaced by $\beta_{j}^{2}$.

6- On page 17, the authors write "one should fix first the value of $\theta$ in Eq. (56) and then consider the large $\ell$ regime". However, Eq. (56) is already assuming the large $\ell$ regime, given that it was produced from the asymptotics of (quasi-)Toeplitz determinants. I assume that for this purpose $\theta$ should be fixed already at the stage of calculating the correlation matrices of Eqs. (23) and (34); do the authors agree?

7- On page 20, the authors changed Eq. (83) following my comment on a discrepancy with Eq. (82), but now the operator that appears in Eq. (83) is not unitary (in the exponent there is a Hermitian operator rather than an anti-Hermitian one). There are several modifications that may solve this issue, and the one the authors choose should match Eq. (82).

8- A small methodological comment regarding the parity argument appearing on page 23. I find the explanation there a little confusing: in general, eigenstates of $H_{XY}$ (or $H_{XY}'$) should not necessarily be eigenstates of $P_{z}$ just because the two operators commute; this is not an algebraic requirement. The reason why the state should be an eigenstate of the parity operator $P_{z}$ is fermionic superselection rules (following the definition of "physical states" for fermions, e.g. in [Banuls, Cirac, and Wolf, Phys. Rev. A 76, 022311]). I would suggest changing this short explanation accordingly.

9- On page 24, there is a wrong sign in the expression for $W_{j}$ right after Eq. (101): the first term in the product should be $I+\Gamma$.

10- On page 25, the time evolution written for $c_{k}\left(t\right)$ before Eq. (115) is incorrect, and should be $c_{k}\left(t\right)=\exp\left(-it\epsilon_{k}\right)c_{k}$. This might have an effect on some time-dependent expressions written in the manuscript, but I do not believe this affects the central results in any way (though it would be good if the authors verified this).

  • validity: good
  • significance: high
  • originality: high
  • clarity: good
  • formatting: good
  • grammar: reasonable

Author:  Sara Murciano  on 2023-06-23  [id 3756]

(in reply to Report 1 on 2023-06-19)

1) We thank the referee for their suggestion. Comparing the asymptotic expressions (69) and (74) for $\Delta S_A^{(n)}$ at $t=0$ and $t=\infty$, which we respectively plot in the left and right panels of Fig. 4, we observe that there are values of $\theta$ for which $\Delta S^{(n)}(t=0)<\Delta S^{(n)}(t\to \infty)$. This implies that the asymmetry does not always drop towards its saturation value after the quench, but it can also increase. Therefore, a phenomenon similar to the quantum Mpemba effect found in the quench from a tilted ferromagnetic state cannot be observed in general when we initiate the system in the tilted Néel state and we quench to the XX spin chain. We would like to mention that we are currently working to provide some analytical explanation of this effect.

2) Our claim refers to Fig. 2 of Ref. 5, where we report the result of the entanglement asymmetry of the tilted ferromagnetic state for finite subsystem size $\ell=10$. In that case, despite the expression in the limit $\ell \to \infty$ is monotonic in $\theta$ for any $n$, the same does not hold as $n\to \infty$ for finite $\ell$. We have repeated the same exercise for the tilted N\'eel state, and we have found that the entanglement asymmetry is monotonic in $\theta$ when $n\to \infty$ even for finite subsystem size. We do not report the computation here since we are mainly interested in the result at $\ell \to \infty$. We thank the referee for the question and we have erased that sentence from the main text.

3)-4)-5) We thank the referee for pointing out these typos, which we have fixed.

6) We agree with the observation done by the referee and we have modified the main text accordingly.

7) We thank the referee for pointing out the inconsistency between Eqs. (82) and (83). This was due to a sign problem in Eq. (83), which we have now fixed. The operator of Eq. (83) is now unitary, and implements the Bogoliubov transformation (82).

8) We agree that the eigenstates of $H_{XY}$ and $H'_{XY}$ are not necessarily eigenstates of $ P_z$, we have rephrased the explanation according to the suggestion of the referee mentioning Ref. [77].

9)-10) We thank the referee for pointing out these typos, which we have fixed.

---

## Round 3 · Author Response

We are submitting a revised version of the manuscript entitled "Lack of symmetry restoration after a quantum
quench: an entanglement asymmetry study".

We would like to thank the editors and the reviewers for their work.

We believe that we have addressed the comments of the Reviewer 2 in the new version of the manuscript. Their comment also helped us in improving the final revised version of the manuscript.

---

## Round 3 · List of Changes

• We modified all typos/misleading/wrong expressions.
  • We have added the result of the entanglement asymmetry for the cat ferromagnetic state.
  • We have included several comments, i.e. after Eq. 37, before Eqs. 46 and 53, at the end of Sec. 4 and 5, a paragraph about the multi-variable saddle-point approximation after Eq. 64, after Eq. 92.
  • We have added 4 appendices (A--D) and modified the Appendix (E).

---

## Round 4 · Referee Report · Anonymous (Referee 4) · 2023-6-25

Report

I am satisfied with the changes that the authors have made and with the explanations they have provided following the last round of review, for which I thank them. I recommend the paper for publication.

---

## Round 4 · Author Response

We thank the referee and the editor for their work and the careful reading of the manuscript. In the new version, we have changed the text and corrected the typos according to the suggestions and comments of the referee.

---

## Round 4 · List of Changes

- Typos fixed;
- References added.

---

## Editorial Decision

published